# Stream flow simulation and verification in ungauged zones by coupling hydrological and hydrodynamic models: a case study of the Poyang Lake ungauged zone

Ling Zhang [1], Jianzhong Lu [1, *], Xiaoling Chen [1, 2], Sabine Sauvage [3], José Miguel Sanchez Perez [3]

[1]State Key Laboratory of Information Engineering in Surveying, Mapping and Remote Sensing, Wuhan University, Wuhan 430079, China

[2]Key Laboratory of Poyang Lake Wetland and Watershed Research, Ministry of Education, Jiangxi Normal University, Nanchang 330022, China

[3]ECOLAB, Université de Toulouse, CNRS, INPT, UPS, 31400 Toulouse, France

* *Correspondence to*: Jianzhong Lu (lujzhong@whu.edu.cn)

**Abstract.** To solve the problem of estimating and verifying streamflow without direct observation data, we estimated streamflow in ungauged zones by coupling a hydrological model with a hydrodynamic model, using the Poyang Lake Basin as a test case. To simulate the streamflow of the ungauged zone, we built a SWAT model for the entire catchment area covering the upstream gauged area and ungauged zone; and then calibrated the SWAT model using the data in the gauged area. To verify the results, we built two hydrodynamic scenarios (the original and adjusted scenarios) for Poyang Lake using the Delft3D model. In the original scenario, the upstream boundary condition is the observed streamflow from the upstream gauged area, while it is the sum of the observed from the gauged area and the simulated from the ungauged zone in the adjusted scenario. The experimental results showed that there are a stronger correlation and lower bias ($R^2 = 0.81$, PBIAS = 10.00%) between the observed and simulated streamflow in the adjusted scenario compared to that ($R^2 = 0.77$, PBIAS = 20.10%) in the original scenario, suggesting the simulated streamflow of the ungauged zone is reasonable. Using this method, we estimated the streamflow of the Poyang Lake ungauged zone as $16.4 \pm 6.2$ billion $m^3$/a, representing ~11.24% of the annual total water yield of the entire watershed. Of the annual water yield, 70% (11.48 billion $m^3$/a) concentrates in the wet season, while 30% (4.92 billion $m^3$/a) comes from the dry season. The ungauged streamflow significantly improves the water balance with the closing error decreased by 13.48 billion $m^3$/a (10.10% of the total annual water resource) from $30.20 \pm 9.1$ billion $m^3$/a (20.10% of the total annual water resource) to $16.72 \pm 8.53$ billion $m^3$/a (10.00% of the total annual water resource). The method can be extended to other lake, river, or ocean basins where observation data is unavailable.

## 1 Introduction

In recent years, floods and droughts have occurred frequently (Cai et al., 2015; Tanoue et al., 2016), threatening lives and health, reducing crop yields and hindering economic development (Lesk et al., 2016; Smith et al., 2014). To reduce the damage to the population, agriculture and economy, we should attempt to predict floods and droughts precisely. However, in

watersheds, there is ungauged zones lack streamflow observations. The ungauged streamflow is difficult to estimate and is usually neglected in water yield estimations, which can result in floods/droughts predictions being not accurate enough.

These ungauged zones are an area of interest in Ungauged Basins (Sivapalan et al., 2003). Ungauged zones, which stretch from the downstream boundary of a gauged basin to the upper boundary of an adjacent water body, exist in river, lake and ocean catchments universally. An ungauged zone usually occupies a large proportion of an entire watershed (Dessie et al., 2015; Li et al., 2014); thus, neglecting ungauged zones adds uncertainty in models of estimating the water yield. In addition, the ungauged zone is usually located in flat topography with a dense river-net, resulting in turbulent flow without a fixed direction. The dense river-net and turbulent flow make it difficult to observe and estimate streamflow in the ungauged zone.

The streamflow simulation in ungauged zones is one area of interest in the Prediction of Ungauged Basins (PUB) research program (Hrachowitz et al., 2013; Sivapalan et al., 2003). In the PUB research program, data acquisition techniques (Hilgersom and Luxemburg, 2012), experimental studies (McMillan et al., 2012; Ali et al., 2012), advanced models and strategies (Harman, 2008), and new hydrological theory (Kleidon et al., 2013) have been developed to improve hydrological prediction results for ungauged zones.

In the PUB research, methods for streamflow prediction in streamflow ungauged zones focus on simple water balance equations and the transformation of hydrological information (Dessie et al., 2015; Song et al., 2015). For simple water balance equations, there are no parameters to be calibrated. Feng et al. (2013) defined streamflow as the difference between precipitation and evapotranspiration. SMEC (2008) determined the streamflow of the ungauged zone based on a lake water balance equation using measured lake water levels and inflow discharges from the upstream gauged catchment. This method is not suitable for accurate streamflow simulation in the ungauged zone.

Some researchers use regionalization methods to simulate streamflow in ungauged zones. The parameters in the gauged areas are calibrated. Then, the parameters are transformed from gauged to ungauged areas. Wale et al. (2009) constructed a regional model for the relationship between the hydrological model parameters and the catchment characteristics. Based on this regional model, the hydrological parameters in the gauged area were transformed to the ungauged zone. However, verification of the ungauged streamflow is not shown in these studies.

Yet other researchers do verification for the ungauged streamflow simulation. Wang et al. (2007) computed the streamflow in an ungauged zone by classifying the underlying surface. The streamflow of each type of surface was calculated based on the surface characteristics. Wang verified the estimation results by comparing the simulated and observed lake water levels. The verification in Ma's study (2011) was based on the water balance of yearly inflow and outflow of the lake. The time resolution is not high enough. Dessie et al. (2015) simulated streamflow in ungauged zones using a rainfall-runoff model and a runoff coefficient. Dessie analyzed the effect of the ungauged zone on the water balance of the lake, which indirectly verified the

streamflow simulation result of the ungauged zone. However, the water balance for indirect verification does not represent the water conservation exactly.

An approach coupling hydrology with hydrodynamics could be used to solve the simulation and verification problems. Usually, a water body (a lake, a river, or an ocean) exists downstream of the ungauged zone. The water body is gauged by streamflow gauging stations at the outlet and water level gauging stations on the water surface. The observations can be used to verify the streamflow simulation result by building a hydrodynamic model for the water body. The method coupling hydrology with hydrodynamic models is widely used to represent the catchment water system and the interaction between catchments and water bodies. Inoue et al. (2008) combined hydrology and hydrodynamic models to simulate the hydrological cycle and hydrodynamic characteristics in a coastal wetland of the Mississippi River Delta, with an effective model performance. Dargahi and Setegn (2011) combined a watershed hydrological (SWAT) model with a 3D hydrodynamic model (GEMSS) to simulate the Tana Lake Basin to address the impact of climate change. Bellos and Tsakiris (2016) combined hydrological and hydrodynamic techniques for flood simulation in the Halandri catchment. However, the method combing a hydrological model and a hydrodynamic model is rarely applied in such ungauged zone. As the ungauged zone is usually located in flat topography with turbulent flow, it is difficult to draw watersheds in the ungauged zone. In addition, allocating the ungauged streamflow to the inflow boundary of a hydrodynamic model is not easy. The ways to drawing watersheds and allocating the streamflow are not mentioned in the  previous studies. The details of coupling hydrology and hydrodynamic models in the ungauged are presented in the study.

The Poyang Lake Ungauged Zone (PLUZ) is a typical example of ungauged zones. The PLUZ is adjacent to Poyang Lake. There are streamflow observations at the outlet of the lake. The streamflow from the PLUZ is usually estimated as the difference between the streamflow at the outlet of the lake and the observed streamflow gauging the upstream area. However, the observations at the outlet of the lake cannot respond to the variation of the watershed hydrology quickly and accurately due to water storage and flood regulation of the lake, which makes the streamflow peak clipped and time-lagged. Traditional method is too coarse for streamflow simulation in the PLUZ.

More attempts have been made for streamflow simulation in the PLUZ. Huang et al. (2011) developed a runoff-flux model especially for the plain area of the PLUZ. The simulation results were verified by comparing observed streamflow at Hukou with the sum of the simulated streamflow in the PLUZ and the gauged streamflow of the gauged upstream in an annual scale. The time scale was coarse. Furthermore, the water storage and flood regulation function of the lake were not taken into consideration. Guo et al. (2011)  simulated the daily runoff of the PLUZ using the Variable Infiltration Capacity (VIC) and multiple-input single-output system (MISO) models. The verification was performed by comparing the simulated results with the estimated results. However, the estimated result was derived from the time-lag equation, so it could not replace the observed value exactly, for the following two reasons:(1) the time-lag equation was a simple hydrodynamic model for the lake, which

is not very accurate; (2) in the equation, the streamflow at Hukou was adjusted by a modified coefficient at the annual scale, which is not reasonable to apply at the daily scale. Most recently, Li et al. (2014) combined the hydrological model (WATLAC) and hydrodynamic model (MIKE), where the streamflow in the ungauged area was calculated by the runoff coefficient. However, there was no verification. In summary, there have been few studies that include effective verification for streamflow simulations in the PLUZ. In the study, the method of combining hydrological and hydrodynamic models is introduced to solve the simulation and verification problem in the PLUZ. Our specific objectives are to: (1) simulate and the verify the streamflow in the PLUZ; (2) analyse the inter-annual and intra annual variations of the ungauged streamflow; (3) analyse the impact of the ungauged streamflow on the lake water balance.

## 2 Study area and data

### 2.1 Study area

Poyang Lake is the largest freshwater lake in China and is connected with the Yangtze River in the north of Jiangxi Province. The catchment is covered by the five major river sub-catchments and the ungauged zone, in Fig. 1(a).

As shown in Fig. 1(a), the Poyang Lake basin includes three parts: the gauged area (the five major river catchments), ungauged zone (the PLUZ) and Poyang Lake. The streamflow of the gauged area was measured by seven streamflow stations (Qiujin, Wanjiabu, Waizhou, Lijiadu, Meigang, Hushan and Dufengkeng). The PLUZ is a plain area and stretches from the seven streamflow stations to the boundary of Poyang Lake. The PLUZ covers an area of 19,867 $km^2$, and amounts to 12% of the lake catchment. The discharges from the gauged area and the PLUZ flow into the lake. Then the water discharges into the Yangtze River at Hukou. The Poyang Lake basin, with an area of 162,000 $km^2$, has a subtropical wet climate characterized by a mean annual precipitation of 1680 mm and annual average temperature of 17.5 ℃. The topography of the Poyang Lake basin varies from upstream hills at an elevation of approximately 2,100 m to downstream plain areas at an elevation of almost 35 m above sea level. The topography of the PLUZ is flat, with a slope of less than five degrees.

The elevation of the lake bed generally decreases from south to the north, with differences of approximately 7 m, as shown in Fig. 1(b). The discharges from the gauged area and the ungauged zone flow into the lake at 11 points ($d_1…d_{11}$). The water level is controlled by the representative stations of Kangshan, Duchang and Xingzi.

### 2.2 Data

We provide data for SWAT and Delft3D models. Data required by the SWAT model include the forcing elements of daily rainfall, evapotranspiration, temperature, relative humidity and wind from 1980 to 2014 collected at 16 national meteorological stations. The stations are distributed uniformly across the area (Fig. 1(a)). These data were downloaded from the China Meteorological Data Sharing Service System (http://data.cma.cn/). The digital elevation model (DEM) of the catchment

originates from SRTM (Shuttle Radar Topography Mission) in 2000. The spatial resolution of the DEM is 90 m. The land-use data were obtained from Landsat TM and ETM+ images in 2000 (Chen el at. 2007). Land-use was categorized into forest (54%), farmland (25%), pasture (10%), water bodies (5%), bare land (3%), urbanization (2%), and wetland (1%). The soil data were generated from HWSD (FAO, 1995). The soil has the following catchment-aggregated proportions: Haplic Acrisols (55%), Cumulic Anthrosols (22%), Humic Acrisols (11%), Haplic Alisols (3%), Haplic Luvisols (2%), and others (7%). The long time series daily discharges at seven gauging stations (Qiujin, Wanjiabu, Waizhou, Lijiadu, Meigang, Dufengkeng, Hushan) from 2000 to 2011 were obtained from the web of hydrological information in Jiangxi. Data required by Delft3D Model included the lake shoreline, topographic data (Qi et al., 2016; Zhang et al., 2015) and hydrological observations. The shoreline was delineated based on the remote sensing image of Poyang Lake during the flood period in 1998, which is the maximum surface area of the lake. The topographic data were measured by the Changjiang Water Resources Commission of China (http://www.cjw.gov.cn). The daily water level at the stations of Xingzi, Duchang and Kangshan, and discharges at Hukou from 2000 to 2011 were downloaded from Jiangxi hydro info website (http://www.jxsw.cn/).

## 3 Methodology

The procedure for the ungauged streamflow simulation and verification, contains three parts (Fig. 2): (1) hydrologic modelling for the Poyang Lake ungauged zone; (2) hydrodynamic modelling for Poyang Lake in two scenarios with or without considering the ungauged streamflow; (3) coupling of hydrological and hydrodynamic models.

In Procedure 1, we built a SWAT model for the entire catchment covering the gauged area and the ungauged zone to simulated streamflow in the PLUZ; and calibrated and validated the SWAT model using the gauged streamflow in the gauged area. In Procedure 2, we built the original and adjusted scenarios for the lake hydrodynamic model to further verify the ungauged streamflow. The original scenario did not take the ungauged streamflow into consideration, unlike the adjusted scenario, which accounted for the ungauged zones. In the adjusted scenario, the hydrological and hydrodynamic modes were coupled. In Procedure 3, we described the coupling of river hydrological and lake hydrodynamic models in details.

In order to analyse the impact of ungauged streamflow on the lake water balance, we described the water balance equation in section 3.4.

### 3.1 Hydrology modelling

We used a SWAT (Soil and Water Assessment Tool) (Arnold et al., 1993) model to simulate streamflow in the PLUZ. SWAT is a physically-based, semi-distributed and river basin-scale hydrological model. It is developed to assess the impact of land management practices on streamflow, sediment and agricultural yields in complex basins with changing soil types, land use and management over long periods of time. For the purpose of modelling, an entire watershed is divided into sub-watersheds

based on rivers and DEM data. Sub-watersheds are portioned into Hydrological Response Units (HRUs), the minimum

research units. Water balance is the driving force of hydrological processes. The hydrological cycle includes two divisions:

runoff-producing on land and flow-routing in channels. The surface runoff volume is calculated using the SCS method (USDA

Soil Conservation Service, 1972). Flow routed through the channel is calculated by the variable storage coefficient method

(Williams et al., 1969). SWAT has already been widely applied to watersheds around the world for streamflow simulation

(Douglas-Mankin et al., 2010;  Arnold et al., 2012; Luo et al., 2016).

A SWAT model should be calibrated and validated by the measured data. The PLUZ is ungauged for streamflow, while there

are streamflow gauging stations (the seven gauging stations) at the upstream boundary of the PLUZ, controlling the upstream

gauged area (Fig. 1(a)). Thus, we established a SWAT model for a larger area, more than just the ungauged zone. The modelled

area covers the upstream gauged area and the ungauged zone (the PLUZ), excluding Poyang Lake (Fig. 1(a)). We use the long

time series of monthly discharges at six gauging stations (Wanjiabu, Waizhou, Lijiadu, Meigang, Dufengkeng and Hushan) to

perform the calibration from 2000 to 2005 and validation from 2006 to 2011. The determination coefficient ($R^2$), Nash-Sutcliffe

efficiency coefficient (Ens), percent bias (PBIAS) and root mean square error (RMSE) are used as the performance indices.

## 3.2 Hydrodynamics modelling

To verify the streamflow simulation results in the PLUZ, we built two hydrodynamic scenarios for the lake using the Delft3D

model. Delft3D simulates the hydrodynamic pattern via the Delft3D-FLOW (Roelvink and van Banning, 1994) module.

Delft3D-FLOW is a multi-dimensional (two-dimension or three-dimension) hydrodynamic and transport simulation program.

The program can calculate unsteady flow by building linear or curvilinear grids suitable for the water boundary, which is

forced by tidal and meteorological data. Delft3D-FLOW is based on the Reynolds-Averaged Navier-Stokes (RANS) equations,

which are simplified for an incompressible fluid under shallow water and Boussinesq assumptions. The RANS equations are

170 solved by the alternative direction implicit finite difference method (ADI) on a spherical or orthogonal curvilinear grid.

Delft3D has ability to simulate water-level variations and flows on surface water bodies in response to forcing elements of

inflow discharges and climate factors, which has been proven by applications on many surface water bodies around the world.

Delft3D is considered appropriate for the wide and shallow characteristics of Poyang Lake.

In the model, the shoreline of lake was delineated as the maximum area of the lake surface to ensure that the dynamic changes

in the lake's surface area did not surpass the inundation area. To better capture the rapid dynamic of inundation area and

minimize the computational effort, the size of the model grids ranged from 200 m to 300 m. The topographic data were

interpolated into each computational node of the model grids. The water level was initialized as the mean of the three

hydrological stations in Poyang Lake on 1 January, 2001, which are Xingzi, Duchang and Kangshan. The corresponding

velocities were initialized as zero. The upper open boundary was set as the upstream discharges. The lower open boundary

was specified as the observed long time series of the daily water level at Hukou station. The model was run from January 1,

2001 to December 31, 2010 and the time step was set as five minutes to meet the Courant-Friedrich-Levy criteria for a stable

condition. The long time series of observed data for water levels at Xingzi, Duchang, and Kangshan gauging stations, and

outflow discharges at Hukou gauging station, were used for calibration from 2001 to 2005 and validation from 2006 to 2010.

Two scenarios were established, the adjusted scenario (Adjusted Scenario) and the original scenario (Original Scenario). We

applies the same hydrodynamic model (Delft3D) in the same study area (Poyang Lake) as the research by Zhang et al (2015).

Therefore, we set the parameter (the Manning roughness coefficient, the eddy viscosity parameter and the critical water depth

for wetting and drying) as the fittest ones calibrated by Zhang et al. for Original Scenario. The parameters in the two scenarios

are set the same.

Original Scenario did not take streamflow in the PLUZ into consideration, unlike Adjusted Scenario, which accounted for the

ungauged zones. In Original Scenario, the upper open boundary was the streamflow from the gauged area, set as the daily

discharges from the seven gauging stations; and there are 9 inflow points—$d_1$, $d_2$, $d_3$, $d_4$, $d_5$, $d_6$, $d_7$, $d_8$ and $d_9$ (Fig. 1b) for the

lake model. In Adjusted Scenario, the upper boundary was the streamflow from the gauged and ungauged areas, set as the sum

of the measured discharges at the seven gauging stations and the simulated streamflow in the PLUZ; and there are 11 inflow

points—*d1, d2, d3, d4, d5, d6, d7, d8, d9, d10,* and *d11* (Fig. 1b) for the lake model. The specific upstream conditions for the

two scenarios are listed in Table 1.

### 3.3 Models coupling

As the ungauged zone is usually in low and flat topography with turbulent flow, it is difficult to draw watersheds in the

ungauged zone. What's more, allocating the streamflow in the ungauged zone to inflow boundary of hydrodynamic model is

not an easy work.

**3.3.1. Drawing the watersheds for the ungauged zone**

The upper boundary condition of the hydrodynamic model in the Adjusted Scenario is the sum of the gauged streamflow from

the gauged area and the simulated streamflow from the ungauged zone (the PLUZ). To determine the upper boundary condition

in Adjusted Scenario, we coupled the hydrological model and hydrodynamic model in space and time.

To make sure the hydrological model and hydrodynamic model were coupled perfectly in space, the delineated sub-basins,

rivers and the outlets of the PLUZ basin should follow the following constraints: (1) The river nets in the PLUZ must be

delineated to link the five major rivers and the inflow points of the lake. (2) The seven gauging stations were set as the outlets

of the gauged basins and the inlets of the PLUZ basin; and the most downstream boundary of the gauged basins should coincide

with the most upstream boundary of the PLUZ basin. (3) The outlets of the PLUZ must completely coincide with the inflow

points of the lake in the hydrodynamic model; and the most downstream boundary of the PLUZ basin should coincide with

the boundary of the lake. (4) The sub-basins of the PLUZ should cover the whole area of the PLUZ. Following the principles, the catchment hydrological model can be seamlessly coupling with the lake hydrodynamic model in space. We first drew the sub-basins, rivers and outlets using the SWAT model. Since the delineated results by the SWAT model may not satisfy these constraints, we edited the rivers, the boundary of sub-basins and the outlets to meet the constraints (Fig. 2).

As shown in Fig. 2, the PLUZ was divided to 14 sub-basins ($b_1, b_2...b_i...b_{14}$), and the ungauged area was divided into 25 sub-

basins ($b_{15}, b_{16}...b_i...b_{39}$). Consequently, 11 outlets of the whole catchment were produced for Adjusted Scenario, coinciding with the lake inflow points—$d_1, d_2, d_3, d_4, d_5, d_6, d_7, d_8, d_9, d_{10}, d_{11}$.

The calibration and validation of the SWAT model was conducted at a monthly scale. However, hydrodynamic model simulation is at a daily scale. To coupling the two models in the same time scale, we use the same parameters of the monthly SWAT model to simulate the ungauged streamflow at the daily scale.

### 3.3.2. Allocating streamflow

To allocate the ungauged streamflow to different inflow points of the lake, the sub-basins were sorted into 11 groups ($group_1,$ $group_2, group_3...group_i...group_{11}$) (Fig. 3). As shown in Fig. 3, the sub-basins in the same group ($gourp_i$) drain to the same inflow point ($d_i$).

Based on the sub-basin groups, we determined the ungauged streamflow gathering to each inflow point of the lake. The

streamflow produced by the PLUZ gathering to $d_i$, is calculated as the difference between the SWAT simulated outflows at the outlets of the whole catchment and the gauged area. The ungauged streamflow contributing to each lake inflow point is listed in Table 2.

In a duration time, water yield can reflect the total amount. So we analysis the water yield variable instead of streamflow. Water yield is computed as the accumulative streamflow in a specific duration. Monthly water yield is the accumulative

streamflow in a specified month. Annual water yield is the accumulative streamflow in a specified year. In the paper, the units of streamflow, monthly water yield and annual water yield are $m^3/s$, $m^3/month$ and $m^3/a$ respectively.

### 3.4 Analysis of lake water balance

In order to analysis the effect of ungauged zone on the lake balance. We construct water balance equations for the lake based on water conservation principles that the difference between of input and output streamflow equals storage change of the lake,

as the follows.

$$Q_{in} + P - E + G + \triangle S + \varepsilon' = Q_{out} \tag{1}$$

where, $Q_{in}$ denotes the inflow from the river basins, P is the precipitation in the lake, $\triangle S$ is the storage change of the lake, and $Q_{out}$ represents the observed outflow at Hukou of the lake. $\varepsilon'$ represents the uncertainties in the water balance, which arise from errors in observed data and other components, such as the ungauged streamflow and model uncertainty. E represents the

evapotranspiration of the lake, less than 2% of the lake outflow. The E data are obtained from Nachang climatology station. G represents the ground water exchange, only 1.3% of the total water balance (Li et al. 2014). Thus, we combine the E, G, and $\varepsilon'$ as the closing error $\varepsilon$. As the summation of $Q_{in}$, P, and $\triangle S$ can be simulated by the hydrodynamic model, the summation is set as the simulated streamflow at Hukou. Traditionally (in Original Scenario), the $Q_{in}$ omits the ungauged streamflow. The water balance equation can be describe as follows.

$$Q_{SimOut,org} + \varepsilon_{org} = Q_{out} \qquad\qquad\qquad (2)$$

where $Q_{SimOut,org}$ represents the simulated streamflow at Hukou from the hydrodynamic model in Original Scenario. $\varepsilon_{org}$ represents the uncertainty of the equation, which arising from the ignorance of the ungauged streamflow, E, G, the error in the observe data, and uncertainty of the hydrodynamic model. As the ungauged zone occupies 12% of the total water balance components (Li et al. 2014), much larger than the other components (E and G, less than 3.3%), the closing error should be

larger than zero on the assumption  that the observe data and hydrodynamic model are sufficient accuracy.

When the ungauged streamflow is taken account (in Adjusted Scenario), the $O_{in}$ contains the gauged and the ungauged streamflow. The water balance equation can be describe as follows.

$$Q_{SimOut,adj} + \varepsilon_{adj} = Q_{out} \qquad\qquad\qquad (3)$$

where $Q_{SimOut,adj}$ represents the simulated streamflow at Hukou from the hydrodynamic model in Adjusted Scenario. $\varepsilon_{adj}$

represents the uncertainty of the equation, which arising from the ignorance of E, G, the error in the observe data, and uncertainty of the hydrodynamic model and the simulated ungauged streamflow result. The partial uncertainties (caused by the ignorance of E, G, the error in the observe data, and uncertainty of the hydrodynamic model) in Adjusted Scenario and Original Scenario are the same. Thus, if the simulated ungauged stream by the SWAT model are sufficient accuracy, the uncertainty in Adjusted Scenario ($\varepsilon_{adj}$) should be smaller than that in Original Scenario ($\varepsilon_{org}$).

**4  Results and discussion**

**4. 1 Calibration and validation of SWAT model and Delft3D model**

To adjust the models to be applied in the Poyang Lake basin availably, we undertook calibration and validation for the SWAT model and the Delft3D model. Table 3 and Fig. 4 show the calibration and validation results for the SWAT model. The observations and simulations at the six gauging stations (Wanjiabu, Waizhou, Lijiadu, Meigang, Hushan and Dufengkeng)

come to a satisfactory agreement, with an $R^2$ or Ens larger than 0.70 and an absolute PBIAS less than 20%, except for Wanjiabu Station. The agreement is also supported by the high consistency between the observations and the simulation in terms of amplitude and phase, although the simulated peak streamflow did not accurately match the observations, producing underestimation  and overestimation (Fig. 4). Nevertheless, the calibration and validation results demonstrate that the SWAT model is generally capable of simulating the streamflow of the catchment.

Table 4 and Fig. 5 show the calibration and validation results for the Delft3D model. The observations and simulations at the four gauging stations (Xingzi, Duchang, Kangshan and Hukou) come to a satisfactory agreement, with an $R^2$ or Ens larger than 0.70 and an absolute PBIAS less than 25%. The agreement is also supported by the high consistency between the observation and simulation, although there are obvious discrepancies during the low water level period (Fig. 5a, Fig. 5b, Fig. 5c) and the highly changed flow velocity period (Fig. 5d). The mismatch probably arises from the decreased elevation of lake

bed from the south to the north and the dynamic variation between wetlands and lake areas. The dynamic variation causes the lake to be a river in dry periods and turn into a lake in flood periods, which is difficult to accurately model. Nonetheless, model calibration and validation results demonstrate that the Delft3D model has the capability to simulate the hydrodynamic characteristics of Poyang Lake.

## 4.2 Streamflow verification in the ungauged zone

To further verify the streamflow simulation results in the ungauged zone, we compared the two hydrodynamic simulation results from Adjusted Scenario and Original Scenario. The Adjusted Scenario took the streamflow in the PLUZ into consideration, while Original Scenario omitted the streamflow in the PLUZ. The hydrodynamic simulation result in Adjusted Scenario is improved compared to the Original Scenario, shown in Table 4 and Fig. 6.

Table 4 shows the results of the two scenarios in two aspects: the lake water level and outflow. For the lake water level, the

absolute PBIAS decreases from 0.85%, 3.18%, and 1.56% in Original Scenario to 0.48%, 2.67%, and 1.21% in the Adjusted Scenario while the $R^2$ keeps the same. The water level simulated result is only a bit improved when inflow to the lake increase by ~10%, due to the large area of the lake. In fact, the simulated water level is already good enough ($R^2 > 0.85$, the absolute of PBIAS < 4%) in Original Scenario. It is not easy to improve the water level simulated result by adding the inflow, only ~10% of the total water resource. However, for the lake outflow discharges, the simulated results in Adjusted Scenario produce a

higher $R^2$ (0.81) and lower absolute PBIAS (10.00 %), compared to that ($R^2 = 0.77$ and PBIAS = 20.01%) in Original Scenario. The evidence suggests an improved simulation result in the Adjusted Scenario when the ungauged streamflow is taken into account, compared to that in Original Scenario when the ungauged streamflow is neglected. The result indicates the ungauged simulated result is reasonable.

Figure 6 show the comparison of the streamflow simulated accuracy in Adjusted Scenario and Original Scenario. The $R^2$ is

larger and RMSE is smaller in Adjusted Scenario than that in Original Scenario during the period from 2001 to 2009. The larger $R^2$ and smaller RMSE indicates a more significant correlation and narrower discrepancy between the simulated and observed streamflow in Adjusted Scenario. The improved simulated result of the hydrodynamic model in Adjusted Scenario indicates that the ungauged simulated streamflow is reasonable. Although in 2010 the simulated result in Adjusted Scenario is not better than that in the Original Scenario (red shadow in the Fig. 6), the opposite result may cause by the dike burst in the

Fuhe basin (Feng et al. 2011) since the SWAT model and Delft3D model don't consider the dike burst. Thus, it doesn't demonstrate the ungauged streamflow is unreasonable in 2010.

**4.3 Stream flows simulation result of the ungauged zone**

We do monthly (Fig. 7) and annual (Fig. 8) statistic for the ungauged streamflow, to study the intra-annual and inter-annual variations. As shown in Fig. 7, monthly water yield of the ungauged zone shows clearly seasonality. In a particular year, the

maximum monthly water yield varies from 1.676 to 7.712 billion $m^3$/month, occurring between April and July (Fig. 7 (a) and (b)); and the minimum monthly water yield varies from 0 to 0.508 billion $m^3$/month, occurring between November and the next February (Fig. 7 (a) and (b)). In the Poyang Lake basin, the precipitation mainly concentrates in the period from March to July (the wet season) and there is less rain during the period from September to next March (the dry season) (Fig. 7 (c)). Nearly 70% of the annual streamflow and nearly 65% of the annual precipitation, come from the wet season. The ungauged

streamflow seasonal variations are consistent with the change of the precipitation, as precipitation is one of the import driving forces for streamflow.

Inter-annual variation is also apparent. Both the month and amount of maximum monthly water yield appear different in different years, as well as that of minimum monthly water yield. For the ten years (2001-2010), the maximum monthly water yield occurred in 2010, when five of twelve month maintained high amount of streamflow (Fig. 7 (a)). Indeed, a flooding event

happened in June 2010 due to the dike burst, causing more than 10 thousand people exposed their lives in danger. The minimum monthly water yield reached the minimum in 2007. In fact, in 2007 Jiangxi province experience severe drought (Feng et al. 2011). The severe flood and drought can also be suggested in Fig. 8. As the water yield is affected by the extreme climate, the long time series of water yields can also reflect flood/drought conditions in Poyang Lake area, in reverse.

Annual streamflow of the ungauged zone shows a clear declining trend (P<0.05, from t-test), at a rate of -1.02 billion $m^3$/a

(dashed line in Fig. 8) during the period from 2001 to 2009. The annual streamflow in the dry and wet season is are decreased by -0.67 billion $m^3$/a and -0.34 billion $m^3$/a respectively from 2001 to 2009. In 2010, the annual streamflow recovered to a high level of 28.07 billion $m^3$/a.

The mean annual water yield in the PLUZ totals 16.4 ±6.2 billion $m^3$, encompassing 11.24% of that from the whole Poyang Lake watershed. The result is close to that from Li's research (Li et al. 2014), where the ungauged streamflow amounts to

~12%. The similar results indicate that the streamflow simulation result of the PLUZ is reasonable. Of the annual water yield, nearly 70% (11.48 billion $m^3$) concentrates in the wet season while 30% (4.92 billion $m^3$) comes from the dry season. Such a great contribution to the inflow of Poyang Lake could have a great effect on the water balance of the lake.

**4.4 The impact of the ungauged zone on the water balance**

In order to analyze the impact of the ungauged streamflow on the lake water balance (seen in section 3.4), we calculate the

closing errors based on the equation 2 and 3: $\varepsilon_{adj}$ when the ungauged streamflow is considered (Adjusted Scenario) and $\varepsilon_{org}$

when the ungauged streamflow is omitted (Original Scenario), in Fig. 9. As shown in Fig. 9, for most months (nearly 83%),

the absolute value of $\varepsilon_{adj}$ is smaller than that of $\varepsilon_{org}$, which can demonstrate the ungauged streamflow improves the lake water

balance.

However, there are some exceptional dot pairs colored in red (outlier, only 17%) in Fig. 9. For the exceptional, the absolute

$\varepsilon_{adj}$ is not less than the absolute $\varepsilon_{org}$ as the above. All the exceptional almost concentrates in the high flow period from July to

October (Fig. 9). That is an unstable stage when backward flow from Yangtze River usually appears and the water level of

Yangtze River usually keeps high (David et al. 2006), which can result in high dynamical changed flow. Thus, more

uncertainties would be added to the measured data and the hydrodynamic model during unstable season (July to October)

compared to the stable season (January to June, December to November). High dynamic changed flow may cause the

streamflow overestimated randomly. High water level of Yangtze River also can leads to overestimated streamflow at Hukou,

compared to the conditions in normal water level. What's more, frequent water abstraction for irrigation from July to October

can also strength the overestimation situation. The accumulative estimation can even lead the closing error less than zero

between July and October (Fig. 9), which is opposite to that the closing error should be more than zero described in section

3.4. The evidence suggests that the hydrodynamic model is not accuracy enough to simulate the streamflow during the unstable

season. During the time, the added input component could make the ever overestimated streamflow larger. Thus, the closing

error will be extended. That's why when $\varepsilon_{org}$ is less than zero, the $\varepsilon_{adj}$ will be more less than zero (the red dot pairs in Fig. 9).

The evidence just demonstrates that the hydrodynamic model is not accuracy enough to simulate the lake input components

during the unstable season from July to October. It doesn't deny the role of ungauged simulated streamflow in improving the

lake water balance.

The ungauged streamflow decreases the annual average closing error of water balance by 13.48 billion $m^3$/a (10.10% of the

total annual water resource) from $30.20 \pm 9.1$ billion $m^3$/a (20.10% of the total annual water resource) to $16.72 \pm 8.53$ billion

$m^3$/a (10.00% of the total annual water resource) for 2001-2010. The evidence also suggests the ungauged simulated

streamflow is reasonable.

**5  Conclusions**

A method coupling hydrology and hydrodynamics can be used to simulate and verify streamflow in ungauged zones, solving

the simulation and verification problems caused by the unavailability of streamflow observations.

The hydrological and hydrodynamic models are coupled seamlessly in both space and time. The method of coupling the models was presented in detail for the first time and was applied in the case study successfully. Using this method, we estimated that the ungauged zone of Poyang Lake produces a streamflow of approximately 16.4 billion m$^3$, representing approximately 11.4% of the total inflow from the entire watershed. The ungauged streamflow significantly improves the water balance with the closing error decreased by 13.48 billion m$^3$/a (10.10% if the total annual water resource) from 30.20 billion m$^3$/a (20.10% of the total annual water resource) to 16.72 billion m$^3$/a (10.00% of the total annual water resource).

The method can be extended to other lake, river, or ocean basins where streamflow observation data are unavailable, producing reasonable streamflow simulation results in ungauged zones. Reliable streamflow simulation results in ungauged zones contribute to more accurate and reliable water yield predictions, which provides a deep understanding of hydrology for hydrological engineers and scientists and helps governments develop better water management plans. Furthermore, this method is an area of interest of Prediction in Ungauged Basins (PUB) and provides streamflow prediction and validation aids in PUB research.

**Data availability**

All data can be accessed as described in Sect. 2.2.

**Acknowledgements**

This work was funded by the National Natural Science Funding of China (NSFC) (Grant No. 41331174), Natural Science Foundation of Hubei Province of China (2015CFB331); the Special Fund by Surveying & Mapping and Geoinformation Research in the Public Interest (201512026), the Collaborative Innovation Center for Major Ecological Security Issues of Jiangxi Province and Monitoring Implementation (Grant No. JXS-EW-08), the Open Foundation of Jiangxi Engineering Research Center of Water Engineering Safety and Resources Efficient Utilization (OF201601), and the LIESMARS special research funding.

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

**Tables**

Table 1. The upstream boundary conditions of the Delft3D model in the Original and Adjusted Scenarios. Od1, Od2, Od3, Od4, Od5, Od6, Od7, Od8, and Od9 represent the streamflow set at *d1*, *d2*, *d3*, *d4*, *d5*, *d6*, *d7*, *d8* and *d9*, respectively, in the Original Scenario. Ad1, Ad2, Ad3, Ad4, Ad5, Ad6, Ad7, Ad8, Ad9, Ad10, and Ad11 represent the streamflow set at *d1*, *d2*, *d3, d4*, *d5*, *d6*, *d7*, *d8* , *d9*, *d10*, and *d11*,

respectively, in the Adjusted Scenario. *B1*, *b2*…and *b13* represent the subbasins in the PLUZ (Fig. 2(b)). $Q_{gau,di}$ and $Q_{ungau,di}$ represent the gauged and ungauged streamflow gathering to the point of *di*, respectively. $Q_{ungau,di}$ will be calculated in the model linking section (seen table 2).

| Scenarios | Inflow Points | Streamflow set at different points |
|---|---|---|
| Original Scenario | *d1* | Od1: the observed streamflow at the Qiujin station ($Q_{gau,d1}$) |
| | *d2* | Od2: 50% of the observed streamflow at the Wanjiabu station ($Q_{gau,d2}$) |
| | *d3* | Od3: 10% of the observed streamflow at the Wanjiabu station ($Q_{gau,d3}$) |
| | *d4* | Od4: 20% of the observed streamflow at the Wanjiabu station ($Q_{gau,d4}$) |
| | *d5* | Od5: 20% of the observed streamflow at the Wanjiabu station ($Q_{gau,d5}$) |
| | *d6* | Od6: the observed streamflow at the Lijiadu station ($Q_{gau,d6}$) |
| | *d7* | Od7: the observed streamflow at the Meigang station ($Q_{gau,d7}$) |
| | *d8* | Od8: the observed streamflow at the Hushan station ($Q_{gau,d8}$) |
| | *d9* | Od9: the observed streamflow at the Dufengkeng station ($Q_{gau,d9}$) |
| Adjusted Scenario | *d1* | Ad1: the summation of $Q_{ungau,d1}$ and $Q_{gau,d1}$ |
| | *d2* | Ad2: the summation of $Q_{ungau,d2}$ and $Q_{gau,d2}$ |
| | *d3* | Ad3: the summation of $Q_{ungau,d3}$ and $Q_{gau,d3}$ |
| | *d4* | Ad4: the summation of $Q_{ungau,d4}$ and $Q_{gau,d4}$ |
| | *d5* | Ad5: the summation of $Q_{ungau,d5}$ and $Q_{gau,d5}$ |
| | *d6* | Ad6: the summation of $Q_{ungau,d6}$ and $Q_{gau,d6}$ |
| | *d7* | Ad7: the summation of $Q_{ungau,d7}$ and $Q_{gau,d7}$ |
| | *d8* | Ad8: the summation of $Q_{ungau,d8}$ and $Q_{gau,d8}$ |
| | *d9* | Ad9: the summation of $Q_{ungau,d9}$ and $Q_{gau,d9}$ |
| | *d10* | Ad10: $Q_{ungau,d10}$ |
| | *d11* | Ad11: $Q_{ungau,d11}$ |

Table 2. The ungauged streamflow allocated to the lake inflow points of the dynamic model in the Adjusted Scenario. $Q_{ungau,di}$ represent the ungauged streamflow gathering to the inflow point of di. d1, d2, d3… d11 are the inflow points in the Delft3D model and the outlets in the SWAT model (Fig. 1(b) and Fig. 3). b1, b2, b3…b11 are the subbasins in the PLUZ (Fig. 3(b)). $Q_{swat,di}$ represent the simulated discharges at the outlet (di) from the SWAT model. $Q_{swat,Qiujin}$, $Q_{swat,Wanjiabu}$, $Q_{swat,Waizhou}$, $Q_{swat,Lijiadu}$, $Q_{swat,Meigang}$, $Q_{swat,Hushan}$, and $Q_{swat,Dufengkeng}$ represent the simulated discharges at the outlets of Qiujin, Waizhou, Lijiadu, Meigang, Hushan and Dufengkeng respectively, from the SWAT model.

| the lake inflow point ($d_i$) | the subbasins draining to $d_i$ | the ungauged streamflow gathering to di |
|---|---|---|
| **d1** | b12, b13 and b14 | $Q_{ungau,d1}$: $Q_{swat,d1}$- $Q_{swat,Qiujin}$- $Q_{swat,Wanjiabu}$ |
| **d2** | b11 | $Q_{ungau,d2}$: $Q_{swat,d2}$- $50\%*Q_{swat,Waizhou}$ |
| **d3** | b10 | $Q_{ungau,d3}$: $Q_{swat,d3}$- $10\%*Q_{swat,Waizhou}$ |
| **d4** | b9 | $Q_{ungau,d4}$: $Q_{swat,d4}$- $20\%*Q_{swat,Waizhou}$ |
| **d5** | b8 | $Q_{ungau,d5}$: $Q_{swat,d5}$- $20\%*Q_{swat,Waizhou}$ |
| **d6** | b7 | $Q_{ungau,d6}$: $Q_{swat,d6}$- $Q_{swat,Lijiadu}$ |
| **d7** | b6 | $Q_{ungau,d7}$: $Q_{swat,d7}$- $Q_{swat,Meigang}$ |
| **d8** | b4 and b5 | $Q_{ungau,d8}$: $Q_{swat,d8}$- $Q_{swat,Hushan}$ |
| **d9** | b3 | $Q_{ungau,d9}$: $Q_{swat,d9}$- $Q_{swat,Dufengkeng}$ |
| **d10** | b2 | $Q_{ungau,d10}$: $Q_{swat,d10}$ |
| **d11** | b1 | $Q_{ungau,d11}$: $Q_{swat,d11}$ |
| **total** | **b1, b2, b3, b4, b5, b6, b7, b8, b9, b10, b11** | **$Q_{ungau,total}$:** **($Q_{swat,d1}$+$Q_{swat,d2}$+$Q_{swat,d3}$+$Q_{swat,d4}$+$Q_{swat,d5}$+$Q_{swat,d6}$+$Q_{swat,d7}$+$Q_{swat,d8}$+$Q_{swat,d9}$+$Q_{swat,d10}$+$Q_{swat,d11}$)-** **($Q_{swat,Qiujin}$+$Q_{swat,Wanjiabu}$+$Q_{swat,Waizhou}$+$Q_{swat,Lijiadu}$+$Q_{swat,Meigang}$+$Q_{swat,Hushan}$+$Q_{swat,Dufengkeng}$)** |


Table 3. Quantitative Assessment of Calibration and Validation for SWAT Model.

| Gauging Station | Index | Model Calibration (2000-2005) | | | Model Validation (2006-2011) | | |
|---|---|---|---|---|---|---|---|
| | | $R^2$ | Ens | PBIAS (%) | $R^2$ | Ens | PBIAS (%) |
| Wanjiabu | monthly discharge | 0.63 | 0.61 | -0.2 | 0.78 | 0.76 | 9.4 |
| Waizhou | monthly discharge | 0.94 | 0.93 | 3.2 | 0.95 | 0.93 | 6.5 |
| Lijiadu | monthly discharge | 0.84 | 0.82 | -9.4 | 0.88 | 0.85 | -16.8 |
| Meigang | monthly discharge | 0.89 | 0.89 | 1.1 | 0.91 | 0.90 | 10.0 |
| Hushan | monthly discharge | 0.81 | 0.78 | 14.2 | 0.76 | 0.75 | 13.9 |
| Dufengkeng | monthly discharge | 0.80 | 0.80 | -4.7 | 0.83 | 0.80 | 9.4 |

Table 4. Quantitative assessment of calibration and validation for streamflow simulation for the Delft3D model.

| Gauging Station | Index | Original Scenario | | | | | | Adjusted Scenario | |
|---|---|---|---|---|---|---|---|---|---|
| | | Calibration (2001-2005) | | Validation (2006-2010) | | All (2001-2010) | | All (2001-2010) | |
| | | $R^2$ | PBIAS (%) | $R^2$ | PBIAS (%) | $R^2$ | PBIAS (%) | $R^2$ | PBIAS (%) |
| Xingzi | Lake water level | 0.99 | 1.2 | 0.99 | 0.45 | 0.99 | 0.85 | 0.99 | 0.48 |

| | | | | | | | | | |
|---|---|---|---|---|---|---|---|---|---|
| Duchang | Lake water level | 0.97 | 4.74 | 0.99 | 2.78 | 0.97 | 3.18 | 0.97 | 2.67 |
| Kangshan | Lake water level | 0.85 | 2.86 | 0.88 | 1.72 | 0.86 | 1.56 | 0.86 | 1.21 |
| Hukou | Lake outflow discharge | 0.75 | 19.46 | 0.80 | 21.47 | 0.77 | 20.10 | 0.81 | 10.00 |

## Figures

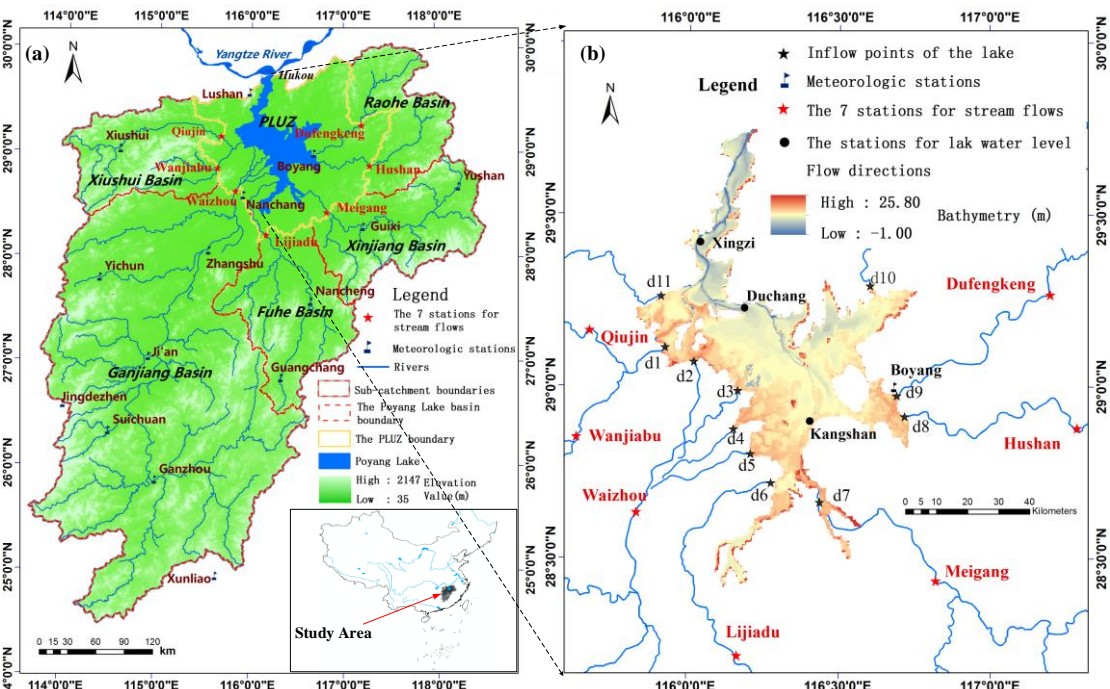

Figure 1. Study area and the related data. (a) The location of the Poyang Lake watershed, PLUZ, five major river sub-catchments, meteorological stations, and hydrological stations; (b) location of the lake, inflow points, and water level stations.

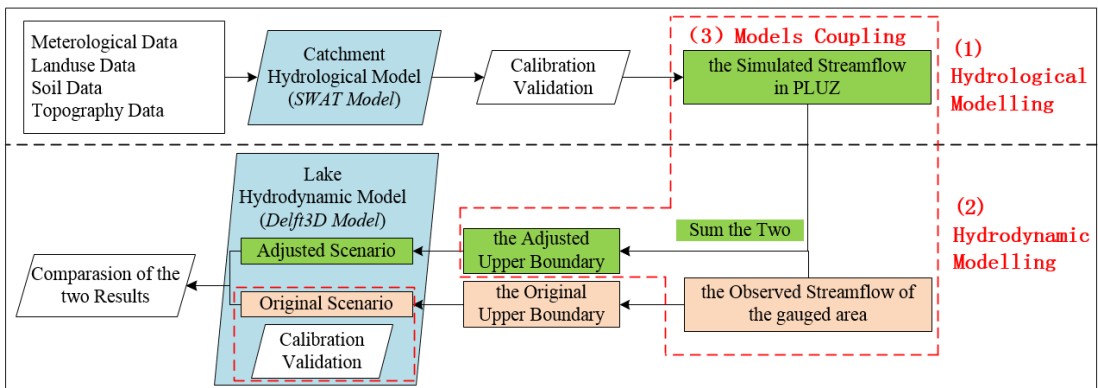

Figure 2. Conceptual flow chart for streamflow simulation and verification in ungauged zones by coupling hydrological and hydrodynamic models. The flow chart includes three parts: Hydrological modelling, Hydrodynamic modelling, Models Coupling.

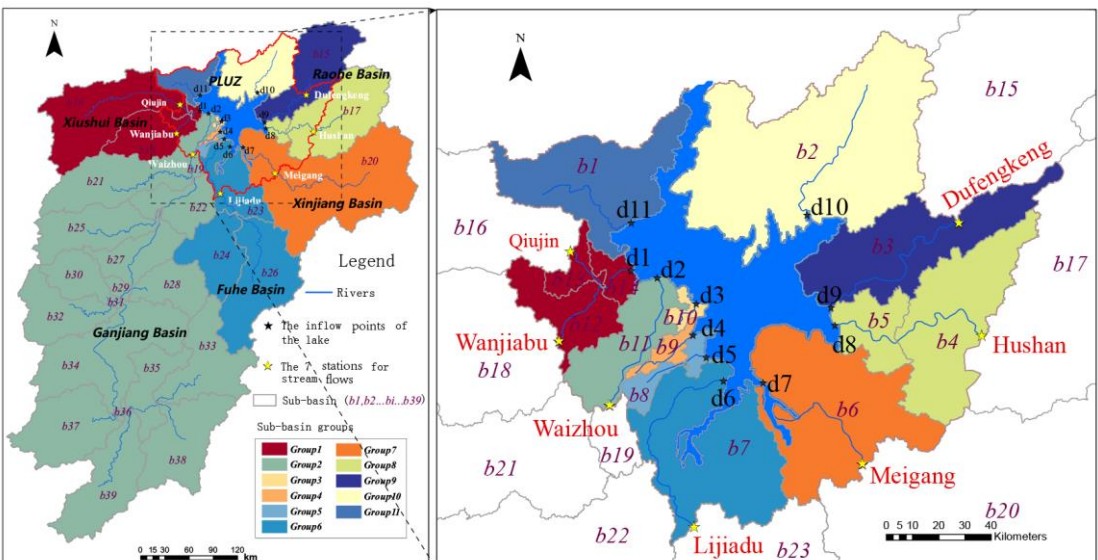

Figure 3. The abridged general view coupling the catchment and lake models in space: (a) streamflow partition scheme from the whole basin to the inflow points ($d_1$, $d_2$...$d_i$...$d_9$) of the lake; (b) streamflow partition scheme from the PLUZ to the inflow points ($d_1$, $d_2$...$d_i$...$d_{11}$) of the lake. The sub-basins in the same group (*group$_i$* colored the same) drains to the same inflow point ($d_i$) of the lake. Specially, in the model, 50%, 30%, 10%, 10% of the streamflow from sub-basins in Ganjiang sub-catchment was set to flow into the lake at points $d_2$, $d_3$, $d_4$, and $d_5$ respectively.

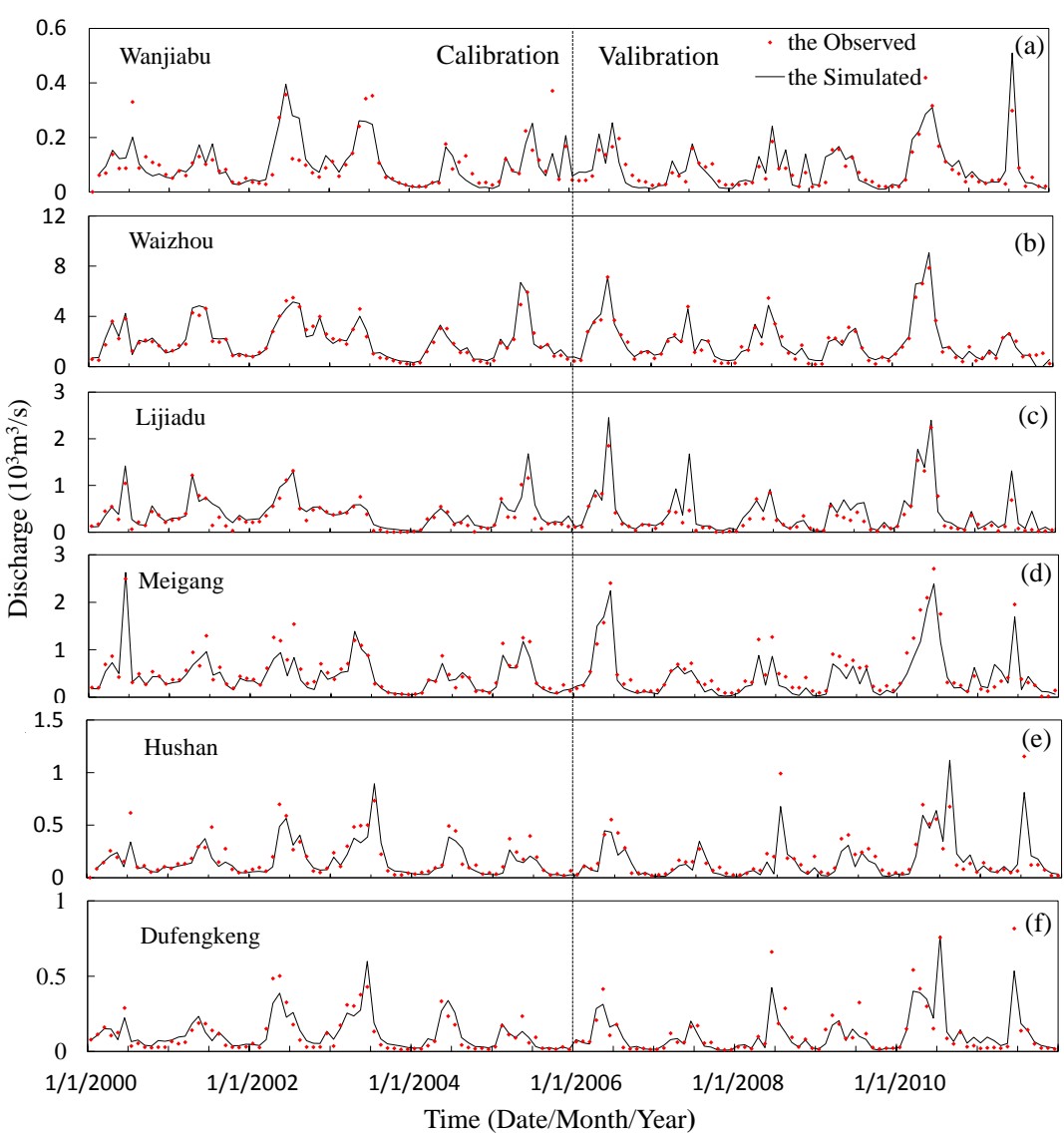

Figure 4. Comparison

of observations and the simulated results by the SWAT Model for calibration (2000-2005) and validation (2006-2011). Subfigures (a), (b), (c), (d), (e) and (f) are the calibration and validation results for stations at Wanjiabu, Waizhou, Lijiadu, Meigang, Hushan, and Dufengkeng, respectively.

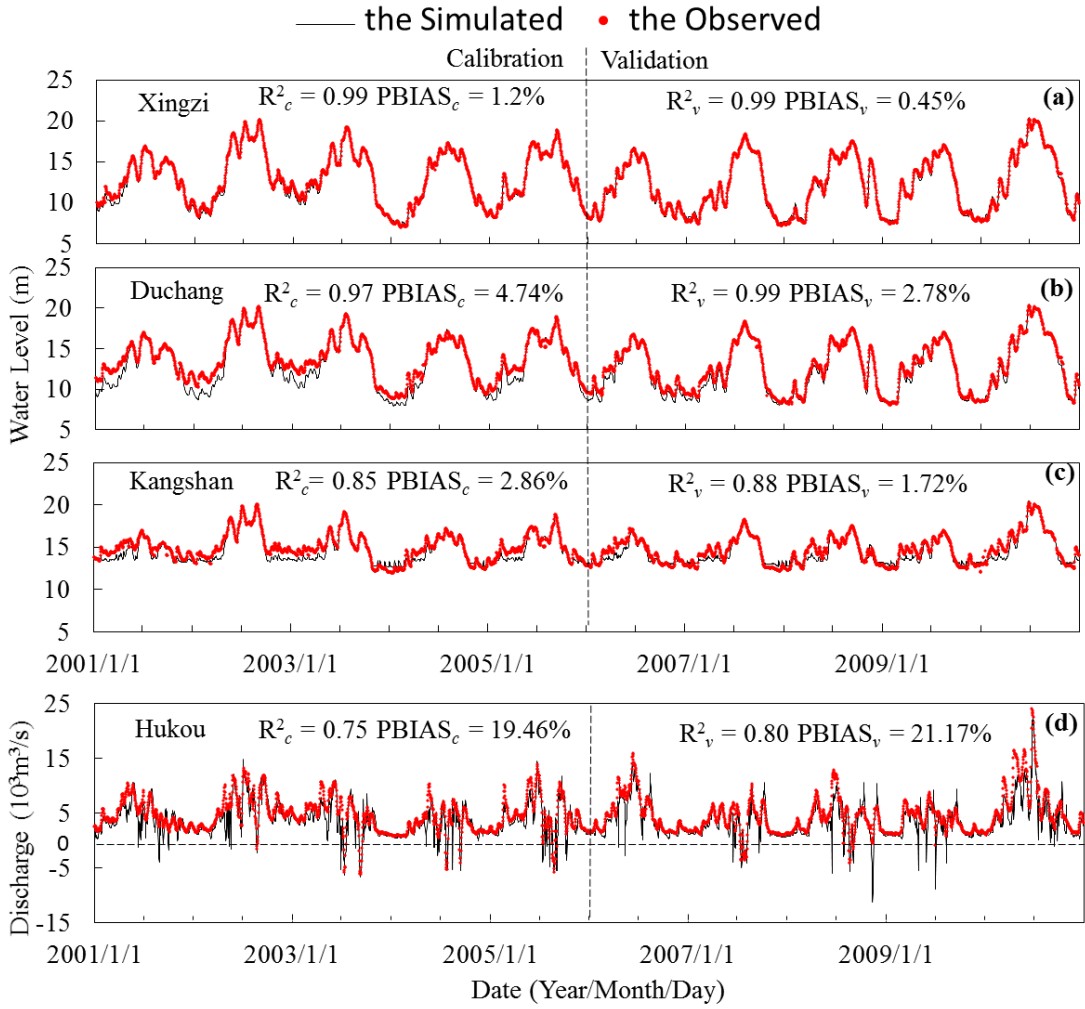

Figure 5. Comparison of the observed (red dotted line) and simulated (black solid line) lake water level at Xingzi, Duchang, and Kangshan stations and outflow discharges at Hukou by the Delft3D Model. The calibration period and validation period are from 2001 to 2005, 2006 to 2010, respectively. $R^2_c$, $PBIAS_c$ and $R^2_v$, $PBIAS_v$ are the calibaration (from 2001-2005) and validation (from 2001-2005) results, respectively.

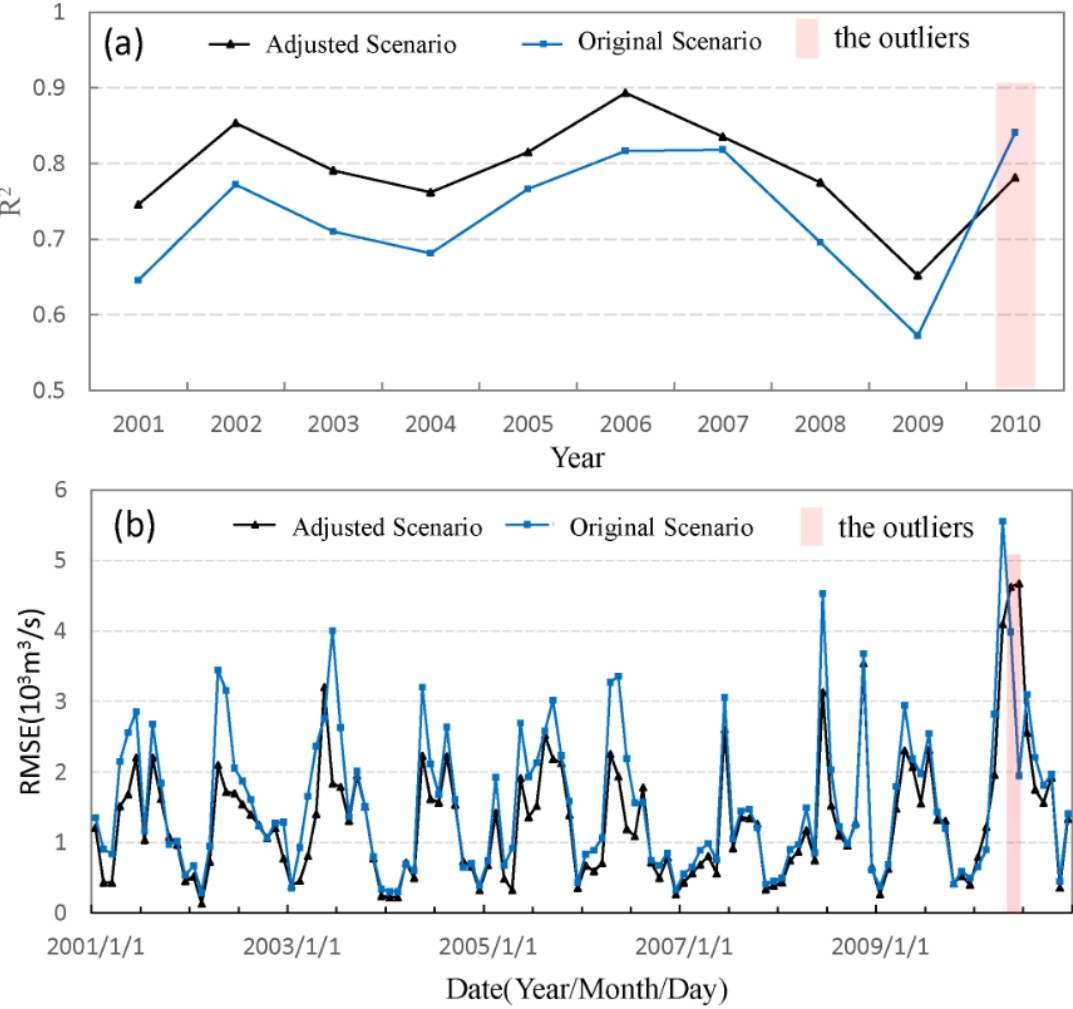

Figure 6. Comparison of the streamflow simulated results at Hukou, in Adjusted Scenario and Original Scenario. The outlier is the data which may affected by the

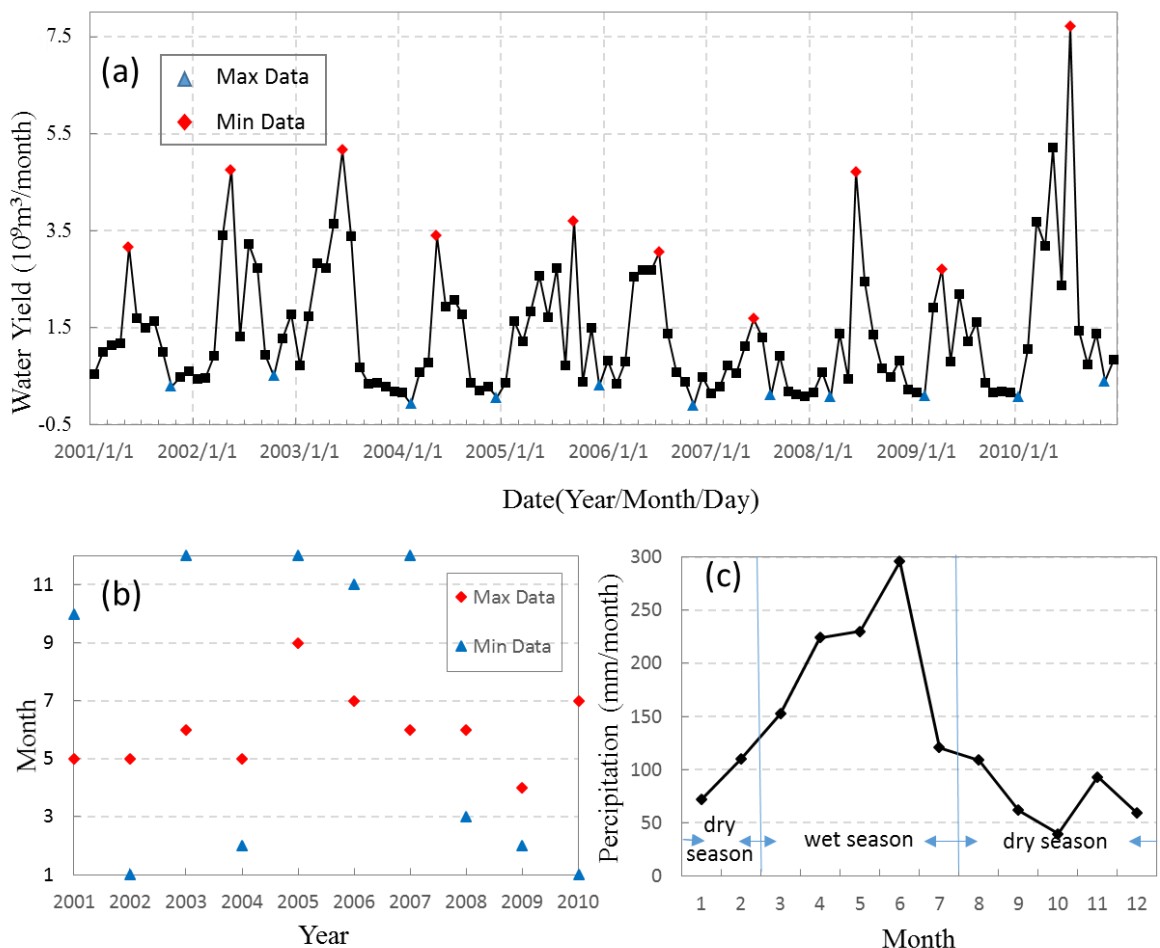

Figure 7. (a)The monthly mean water yield for each month from January 2001 to December 2010 produced by the PLUZ; (b)Maximum and

minimum water yield month distributed from 2001to 2010; (c) The mean monthly precipitation from 2001 to 2010 at Nanchang

meteorological station derived from China's meteorological nets. Max Data and Min Data represent the monthly maximum water yield and

monthly minimum water yield in the particular year respectively.

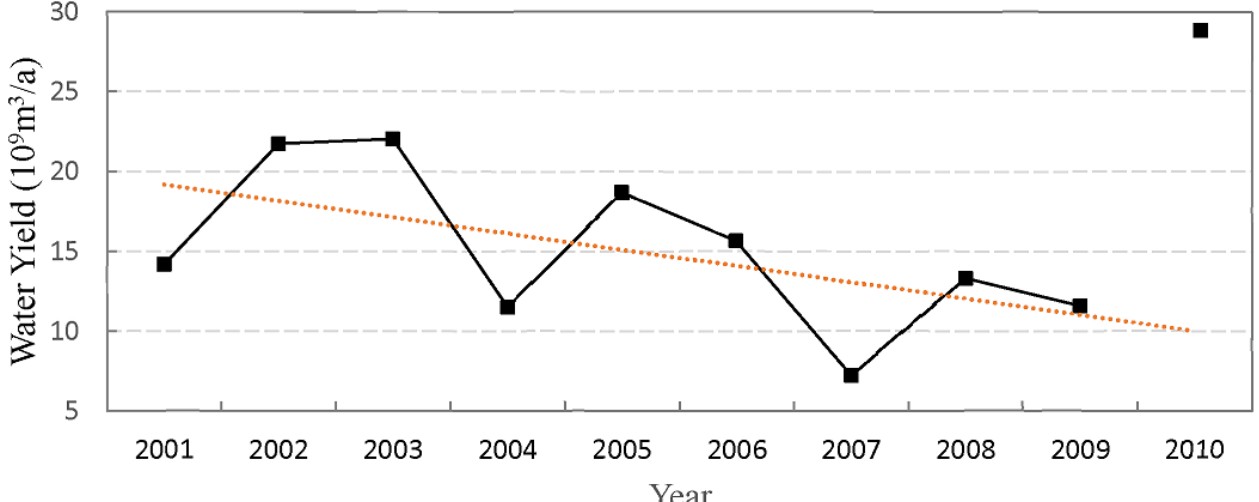

Figure 8. The variation trend of the annual water yield of the ungauged zone from 2001 to 2009. It shows declining trend at a rate -1.02 of

billion m3/a (P < 0.05).

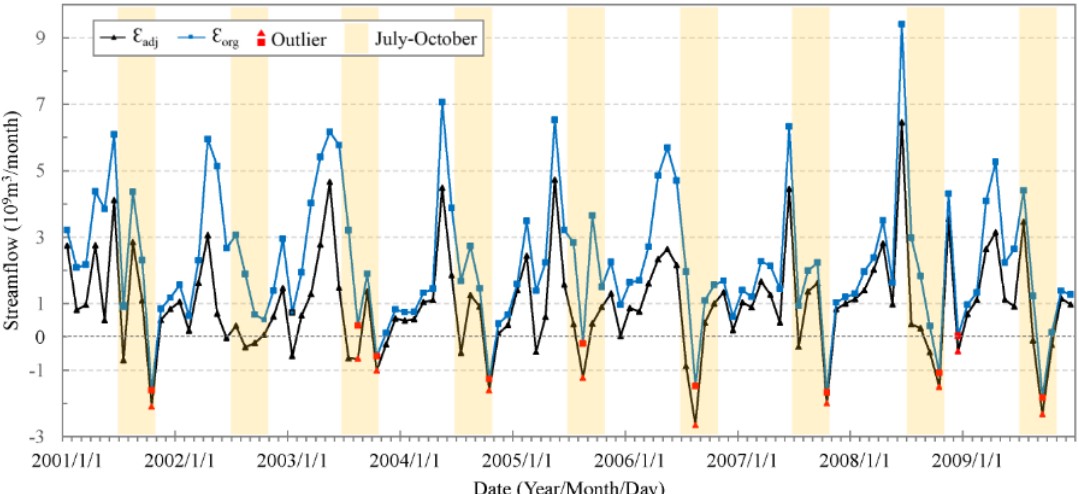

Figure 9. Closing errors of lake water balance: $\varepsilon_{org}$ and $\varepsilon_{adj}$. $\varepsilon_{org}$ is the closing error when the ungauged streamflow is considered and $\varepsilon_{adj}$ is

the closing error when the ungauged streamflow is omitted. Outliers are the point pairs, of which the $\varepsilon_{org}$ is samll than $\varepsilon_{adj}$, expected abnormal.
