# Peer review of "Stream flow simulation and verification in ungauged zones by coupling hydrological and hydrodynamic models: a case study of the Poyang Lake ungauged zone"

_Hydrology and Earth System Sciences, 2017_

## Referee Comment (RC1) · Anonymous Referee #1 · 8 Apr 2017

General comment The manuscript tried to link SWAT with Delft3D to estimate the streamflows in ungagged zones with a case study of the Poyang Lake basin. The topic and the methods sounds interesting but the poor writing make it hard to be understood well. I also think the study might be a reference for other areas. However, the manuscript was not well written and also there are some issues that need to be addressed carefully before it can be considered for publication.

Specific comments 1. The abstract was not well written. (1) 'To estimate streamflow without observation, the authors extend existing techniques . . .', but it is not clear what

is the existing one and what is extended one? A simple 'coupling hydrological model with a hydrodynamic model' is not far clear. (2) L13-15: It is hard to understand. What is land covered area? (3) L15-17: I still did not get what the original and adjusted scenarios are. (4) it is not that convincing to say R2 with higher values and bias with lower values, it would be better to use numbers or a range (e.g., 0.7∼0.8). 2. L29-33: it does not read well, the connection seems not logical. 3. L66-67: What does 'Usually, there are stream flow observation at the lower boundary of the ungauged zone.' Mean? 4. L72-75: Dargahi and Setegn combined a hydrological model (SWAT) with a 3D hydrodynamic model (GEMSS) .... Bellos and Tsakiris . . .. However, . . . there is no clear and specific method of coupling hydrological and hydrodynamic models in space and time.' It is really hard for readers to get what problems or drawbacks others have, and what the novelty of the authors' method is. 5. L101-103: 'We established . . . model was established to . . .' Grammar issue. 6. L103-106: It is strange the end of Introduction was repeating the abstract. 7. L121-124: It reads awkward, and it seems SWAT doesn't need temperature? Were all the data downloaded from Jiangxi hydro info website? 8. Methodology section is too short and lack details. 9. SWAT and Delft3D are the two major approaches of the study; however, there was no description of the two models. 10. L146-147: to simulated ? 11. The results and discussion seems just result description and no discussion was provided. 12. There are many grammar issues here and there, and I believe they need a professional editing service before resubmission.

---

## Referee Comment (RC2) · Anonymous Referee #2 · 9 Apr 2017

This manuscript coupled the hydrological (i.e., SWAT) and hydrodynamic model (i.e., Delft3D) models to verify the simulation of ungagged streamflow taking the Poyang Lake as a case study. The streamflow simulation for ungagged zone is a hot topic in hydrology and the coupling of hydrological model and hydrodynamic modeling is interesting. However, the writing is very poor. Most of sentences are too awkward to be understood even though the grammar of the sentence is correct. After reading the whole of methodology, I could not proceed with the rest of the manuscript, not only because of the poor English writing but also because of the confusing, conflict, and unclear technical details. Based on my understanding, the methods in this study

should be very clear: authors firstly simulated the streamflow of inlets for Poyang Lake using SWAT model; the parameters were calibrated using the observed streamflow from gauges located in the upper streams; then, the simulated streamflow were used as the inflows for hydrodynamic model to simulate the water level and other hydrodynamic characteristics of Poyang Lake; finally, modeled water level and discharge in outlet (Hukou) with and without SWAT simulated inflows were compared. If my understanding is correct, I don't understand why authors used more than five pages to describe this simple procedure and the procedure hasn't been clarified ultimately. For example, I didn't see any coupling of hydrological and hydrodynamic models in sections 3.3 but a lot of confusing water balance equations (i.e., Equation (2-4)), especially for equation (3). Where is the water level change (i.e., surface water storage) in equation (3)? If the water level change is too small to be negligible authors may need to verify it and clarify it in the manuscript. In my opinion, all the Equation (2-4) can be written in one if considering the Poyang Lake as the control volume:

$$Q_{Hukou} = P + Q_{inflows} - E - \Delta SWS_{simulated}$$

where $Q_{Hukou}$ is the discharge in the outlet of Poyang lake; $P$ is the precipitating in Poyang Lake; $Q_{inflows}$ is the summation of the streamflow in all inlets of Poyang Lake; $E$ is the evaporation in Poyang Lake; $\Delta SWS_{simulated}$ is the water level changes in Poyang Lake. Given the authors' methodology is correct, I have another two concerns: has the hydrodynamic model been calibrated? If yes, the bias and error of the simulated ungagged streamflow can be corrected during the calibration of hydrodynamic model which means the verification of simulated streamflow in ungagged zone may ben spurious; the other concern is I don't think it's necessary to use SWAT simulation as the results shown in figure 4 since the discrepancy of two scenarios with and without SWAT is relatively small which may be smaller than the uncertainties in SWAT simulations as shown in Figure 3.

In my opinion, there are serious technical problems with this work. Hence, the results section is likely to be flawed. For this reasons, I couldn't proceed further in the

manuscript as I consider the manuscript unsuitable for publication in HESS, at least, in the current version. My specific comments have been listed below:

1. Line 15: Is the water covered area of the ungauged zone the Poyang Lake? If yes, please revise or it is very confusing. 2. Line 18: how do you conclude "narrower discrepancy"? Please provide some quantification. The same for Line 23 "higher value" 3. Line 29-30: Please rewrite this sentence. 4. Line 40: Please delete the second "stream flow". 5. Line 51-52: Please revise this sentence. Awkward. 6. Line 60: Where is the citation for Ma's study? 7. Line 124: Please provide the spatial resolution for DEM. 8. Line 131: Please provide some examples about the topographic data. 9. Line 134-135: Please provide the temporal scales for water level and discharge. 10. Line 142: What does "sing" mean here? 11. Line 163: Where is the water level change (i.e., surface water storage change) in equation (1) and (3)? If the water level change can be negligible please verify it and clarify it. 12. I didn't read the Result section word by word please carefully read it and revise it based on the revised methodology. 13. Line 440: Please delete Table 3. 14. Line 444: Please switch the Figure 1a and 1b; put figure 1a in left hand side; delete the "Meteorological stations" in legend of Figure 1b since there is no meteorological station; add the scale bar in Figure 1b. 15. Line 450: Figure 2 is confusing, and please revise it. 16. Please provide the line number for figure 3 and also the captions for the subfigures. 17. Based on the figure 4, I don't think it's necessary to use SWAT simulation. The discrepancy of two scenarios is relatively small which may be smaller than the uncertainties in SWAT model as shown in Figure 3. 18. Please delete the figures 5-7.

---

## Author Comment (AC1) · 23 May 2017

We are very grateful to the reviewer for reading the manuscript extremely carefully and forwarding the valuable suggestions for improvement. Point-by-point responses to the reviewers' comments are listed below.

1. General Comments

The reviewer's comment 1: However, the writing is very poor. Most of sentences are too awkward to be understood even though the grammar of the sentence is correct. After reading the whole of methodology, I could not proceed with the rest of the manuscript,

not only because of the poor English writing but also because of the confusing, conflict, and unclear technical details.

The authors' Answer: Thanks for the kind advice. We have invited a professional organization to modify the language. The confusing, conflict, and unclear technical details will be stated clearly.

The reviewer's comment 2: Based on my understanding, the methods in this study should be very clear: authors firstly simulated the streamflow of inlets for Poyang Lake using SWAT model; the parameters were calibrated using the observed streamflow from gauges located in the upper streams; then, the simulated streamflow were used as the inflows for hydrodynamic model to simulate the water level and other hydrodynamic characteristics of Poyang Lake; finally, modeled water level and discharge in outlet (Hukou) with and without SWAT simulated inflows were compared. If my understanding is correct. . .

The authors' Answer: The reviewer is almost right. The procedures of the manuscript are as follows. Procedure 1: we first calculated the streamflow produced by the Poyang Lake ungauged zone (PLUZ). Procedure 2: the modeled water level and discharge in outlet (Hukou) with and without PLUZ streamflow were compared. For Procedure 1, the Poyang Lake ungauged zone including two parts. One is the land cover area of PLUZ, the other one is water covered area (It has been revised as Poyang Lake). The water covered area is Poyang Lake. So in the next step, we calculate the streamflow for the land covered area and Poyang lake separately. The streamflow for the land covered are of PLUZ is simulated by SWAT model. The parameters were calibrated using the observed streamflow from gauges located in the upper streams. The stream-flow in the land coved area of PLUZ is calculated as the difference value of simulated streamflow at inlets of Poyang Lake and the observed streamflow of the upper stream. The streamflow for the Poyang Lake is calculated by a simplified equation (Eq (1)). As the reviewer comments, the equation is not serious for not considering the lake storage change (This will be discussed in the reviewer's comment 4 in General Comments).

[Figure]

The steamflow of PLUZ is the summation of streamflow produced by the land coved area and Poyang Lake. For Procedure 2, the calculated streamflow in PLUZ were used as part of the inflows for hydrodynamic model to simulate the water level and other hydrodynamic characteristics of Poyang Lake. In this processing, two lake hydrodynamic scenarios (the Adjusted Scenario, the Original Scenario) are built. In Adjusted Scenario, inflows for hydrodynamic model is the summation of simulated streamflow in PLUZ and the observed streamflow from gauges located in the upper streams. In Original Scenario, inflows for hydrodynamic model is the observed streamflow from gauges located in the upper streams. The modeled results (water level and discharge of outlet (Hukou)) in Adjusted Scenario, Original Scenario were compared.

The reviewer's comment 3: ...I don't understand why authors used more than five pages to describe this simple procedure and the procedure hasn't been clarified ultimately. For example, I didn't see any coupling of hydrological and hydrodynamic models in sections 3.3

The authors' Answer: Thank you for the valuable suggestion. The coupling is in L197-252. The writing of manuscript may confuse you. In the manuscript, it is loosen coupling. In the Adjusted Scenario, the output of SWAT model is the input of Delft3D model. The streamflow of PLUZ is used as part of the upper inflows for hydrodynamic model. In the Original Scenario when the streamflow of PLUZ is not considered, there are 9 inflow points (d1, d2, d3, d4, d5, d6, d7, d8 and d9) for the lake (Figure 1(b)). Inflows to the points comes from the 7 gauging stations (Qiujin, Wanjiabu, Waizhou, Lijiadu, Meigang, Hushan and Dufengkeng). In the Adjusted Scenario, we should solve the problem: how to allocate the ungauged streamflow in to different inflow points. As the ungauged zone is usually in flat topography with turbulent flow, it is difficult to draw watersheds in the ungauged zone. What's more, allocating the streamflow in the ungauged zone to inflow boundary of hydrodynamic model is not an easy work. The the coupling sections describe the content: how to drawing watersheds in the ungauged zone and allocating the ungauged streamflow to the lake model properly. The section

may confuse you. We will reorganize the coupling section.

The reviewer's comment 4: ...but a lot of confusing water balance equations (i.e., Equation (2-4)), especially for equation (3). Where is the water level change (i.e., surface water storage) in equation (3)? If the water level change is too small to be negligible authors may need to verify it and clarify it in the manuscript. In my opinion, all the Equation (2-4) can be written in one if considering the Poyang Lake as the control volume: $QHukou = P + Qinflows - E - \Delta SWSsimulated$ where QHukou is the discharge in the outlet of Poyang lake; P is the precipitating in Poyang Lake; Qinflows is the summation of the streamflow in all inlets of Poyang Lake; E is the evaporation in Poyang Lake; $\Delta SWSsimulated$ is the water level changes in Poyang Lake.

The authors' Answer: Thank you very for the valuable suggestion. The writing of manuscript may confuse you. In Section 3.3 (Equation (2-4)), we intended to calculated the inflow at different inflow points of the lake in the Adjusted Scenario. The inflow at the point was the summation of three parts: the streamflow in land covered PLUZ, the streamflow in Poyang Lake, and the streamflow from the upper gauged stream (the 7 gauging stations: Qiujin, Wanjiabu, Waizhou, Lijiadu, Meigang, Dufengkeng and Hushan). However, originally I was confused by the concept of STREAMFLOW in Poyang Lake. The streamflow in Poyang Lake should take the lake level change into consideration. However, the lake streamflow is estimated by the lake hydrodynamic model. In the hydrodynamic model, we modeled the changed lake level and streamflow in long time series. The streamflow of the lake should not be the upper inflow boundary of the Lake hydrodynamic model. So we revise the inflow to the lake inflow point as the summation of two parts: the streamflow in land covered area of PLUZ, and the streamflow from the upper gauged stream (the 7 gauging stations). As the lake hydrodynamic model can model the changed lake level and streamflow, we have decided not to take Poyang Lake as part of the ungauged zone. So the PLUZ is redefined the area inside the yellow line and outside the boundary of Poyang Lake (Figure 1(a)).

The reviewer's comment 5: ...Given the authors' methodology is correct, I have another two concerns: has the hydrodynamic model been calibrated? If yes, the bias and error of the simulated ungauged streamflow can be corrected during the calibration of hydrodynamic model which means the verification of simulated streamflow in ungagged zone may be spurious;

The authors' Answer: Thank you very for the valuable suggestion. The writing of manuscript may make you confused. We construct two Scenarios (the Adjusted Scenario, the Original Scenario) for the lake hydrodynamic model. The Adjusted Scenario take the ungauged streamflow into consideration while the Original Scenario does not. The model in the Original Scenario has been calibrated and validated. The model in the Adjusted Scenario use the same parameter as that in the Original Scenario.

The reviewer's comment 6: the other concern is I don't think it's necessary to use SWAT simulation as the results shown in Figure 4 since the discrepancy of two scenarios with and without SWAT is relatively small which may be smaller than the uncertainties in SWAT simulations as shown in Figure 3.

The authors' Answer: Thank you for the comments. The writing may confuse you. If the writing is more clear, your comments is as follows: the other concern is I don't think it's necessary to use SWAT simulation as the results shown in Figure 4 since the discrepancy of two scenarios with and without ungauged streamflow is relatively small which may be smaller than the uncertainties in SWAT simulations as shown in Figure 3. The SWAT and Delft3D are two different models and applied in different specific fields. The acceptable simulation accuracies are different too. In general, SWAT model simulation can be judged as satisfactory if Ens (Nash-Sutcliffe efficiency) > 0.50 and -25% < PBIAS (percent bias) < 25% for streamflow (Van Liew et al. 2007), while for water level (or tide level) simulation by Delft3D model Ens can be larger than 0.90 (some can reach 0.99) and the absolute value of PBIAS can be smaller than 5% (Qi et al., 2016; Zhang et al., 2015). The Delft3D model is steady and the uncertainty is relatively small. Therefore, it is not easy to improve the simulation result of Delft3D model. If the simulated accuracy of Delft3D model can be increased, the increased range should

be small and may be smaller than the uncertainty of SWAT model. However, it is valuable to improve the Delft3D model, although the increased range is small. In the previous research for Poyang Lake dynamic model, the simulated streamflow at the outlet (Hukou) is smaller than the observed in average (Qi et al., 2016; Zhang et al., 2015). This may result from not taking the ungauged streamflow into consideration. So it is valuable and important to test if ungauged streamflow can increase the Poyang Lake dynamic model. And we are curious what the results would be. It is important to get the result: the ungauged zone improves the Delft3D model accuracy although the increase range of the accuracy is small. You comment means the discrepancy should be bigger than the SWAT uncertainty if we want to validate the simulation result of the ungauged streamflow by Delft3D model. That is because the increased discrepancy may be caused by the uncertainty of the SWAT model? If my understanding is right, what we should do is to demonstrate that the discrepancy (the increased accuracy) is not the random fluctuation result of the SWAT uncertainty. We separate the 10 years data into 10 segment. Each segment has one year data. If we can get the discrepancy (the increased accuracy) in all the 10 segment, then we can demonstrate that the discrepancy (the increased accuracy) is not the random fluctuation result. Then the SWAT simulation result can be valid. We are trying to redo the experiment.

2. Specific Comments

The reviewer's comment 1: Line 15: Is the water covered area of the ungauged zone the Poyang Lake? If yes, please revise or it is very confusing.

The authors' Answer: Yes. It will been revised in the manuscript.

The reviewer's comment 2: Line 18: how do you conclude "narrower discrepancy"? Please provide some quantification. The same for Line 23 "higher value"

The authors' Answer: Yes. It has been revised in the manuscript as flows. Experimental results show there was a narrower discrepancy (R2=0.81, PBIAS=10.00%) between the stream flows observed at the outlet of the lake and the simulated stream flows in

adjusted scenarioïijŇcompared to that in original scenario (R2=0.77, PBIAS=20.10%). Using our technique, we estimated that the ungauged zone of Poyang Lake produces stream flows of approximately 180 billion m3; representing about 11.4% of the total inflow from the entire watershed. We also analyzed the impact of the stream flows in ungauged zone on the water balance between inflow and outflow of the lake. These results, incorporating the estimated stream flow in ungauged zone, significantly improved the water balance as indicated by R2 with higher value (0.81) and percent bias with lower value (10.00 %), as compared to the results when the stream flows in the ungauged zone were not taken into account, R2 with lower value (0.41) and percent bias with higher value (18.88%). The method can be extended to other lake, river, or ocean basins where observation data is unavailable.

The reviewer's comment 3: Line 29-30: Please rewrite this sentence.

The authors' Answer: The sentence is revised as follows. In order to reduce the damage to the population, agriculture and economy, we should predict floods and droughts precisely. However, in watersheds there is an ungauged zone lacking stream flow observations. The streamflow of ungauged zone is difficult to estimate, which makes ungauged zones neglected in water yield estimation. Therefore, it is important to estimation the streamflow in the ungauged zone.

The reviewer's comment 4: Line 40: Please delete the second "stream flow".

The authors' Answer: It has been delete.

The reviewer's comment 5: Line 51-52: Please revise this sentence. Awkward.

The authors' Answer: Some researchers use regionalization methods to simulate streamflow in ungauged zones. The parameters in the gauged areas are calibrated. Then the parameters are transformed from gauged to ungauged areas.

The reviewer's comment 6: Line 60: Where is the citation for Ma's study?

The authors' Answer: It is an article in Chinese. The reference is as follows: Ma,

[Figure]

X., Liu, D.: Modeling of interval runoff in the region of Dongting Lake[J]. Journal of Hydroelectric Engineering, 30(5):10-15, 2011.

The reviewer's comment 7: Line 124: Please provide the spatial resolution for DEM.

The authors' Answer: The spatial resolution for DEM is 90 m.

The reviewer's comment 8: Line 131: Please provide some examples about the topographic data.

The authors' Answer: The following examples of topographic data have been added. The reference is as follows: Qi, H., Lu, J., Chen, X., et al. Water age prediction and its potential impacts on water quality using a hydrodynamic model for Poyang Lake, China. Environmental Science and Pollution Research, doi:10.1007/s11356-016-6516-5, 23(13):13327-13341, 2016. Zhang, P., Lu, J., Feng L., et al. Hydrodynamic and inundation modeling of China's largest freshwater lake aided by remote sensing data. Remote Sensing, doi:10.3390/rs70404858, 7(4): 4858-4879, 2015.

The reviewer's comment 9: Line 134-135: Please provide the temporal scales for water level and discharge.

The authors' Answer: Daily scale. The sentence has been revised as follows: The daily observation for water level (at stations of Xingzi, Duchang and Kangshan), and outflow discharges (at Hukou) from 2000 to 2011 were got from Web of hydrological information in Jiangxi.

The reviewer's comment 10: Line 142: What does "sing" mean here?

The authors' Answer: It is a spelling mistake. It should be "using".

The reviewer's comment 11: Line 163: Where is the water level change (i.e., surface water storage change) in equation (1) and (3)? If the water level change can be negligible please verify it and clarify it. (???)

The authors' Answer: The lake water level change has been taken into consideration

in the lake hydrodynamic model. As there are observed stations gauging the water level and lake hydrodynamic model can model the streamflow in Poyang Lake, Poyang Lake is not considered as ungauged zone. (more details in Comment 4 in General Comments)

The reviewer's comment 12: I didn't read the Result section word by word please carefully read it and revise it based on the revised methodology.

The authors' Answer: Thank you for the suggestion. We will modify it based on the the methodology.

The reviewer's comment 13: Line 440: Please delete Table 3.

The authors' Answer: We delete Table 3 in the revised version.

The reviewer's comment 14: Line 444: Please switch the Figure 1a and 1b; put figure 1a in left hand side; delete the "Meteorological stations" in legend of Figure 1b since there is no meteorological station; add the scale bar in Figure 1b.

The authors' Answer: There is a meteorological station of Boyang. And the rest will be revised in Figure 1.

The reviewer's comment 15: Line 450: Figure 2 is confusing, and please revise it. (???)

The authors' Answer: It will be revise as the comment (Figure2).

The reviewer's comment 16: Please provide the line number for figure 3 and also the captions for the subfigures.

The authors' Answer: We will provide the line number for Figure 3. The added captions is as follows: Subfigures (a),(b),(c),(d),(e) and (f) are the calibration and validation result for stations at Wanjiabu, Waizhou, Lijiadu, Meigang, Hushan, and Dufengkeng separately.

The reviewer's comment 17: Based on Figure 4, I don't think it's necessary to use

SWAT simulation. The discrepancy of two scenarios is relatively small which may be smaller than the uncertainties in SWAT model as shown in Figure 3.

The authors' Answer: We are trying to make more analysis (more details in the reviewer's comment 6 in General Comments).

The reviewer's comment 18: Please delete the figures 5-7.

The authors' Answer: We delete the figures 6-7. Figure 5 shows the long time series of ungauged water yield, which is the simulation result of the ungauged zone. So I think it may be valuable.

―――――――――――――――

---

## Author Comment (AC2) · 23 May 2017

We are very grateful to the reviewers for reading the manuscript extremely carefully and forwarding the valuable suggestions for improvement. Point-by-point responses to the reviewers' comments are listed below.

1. General comment

The reviewer's comment 1: . . .but the poor writing make it hard to be understood well. . . .However, the manuscript was not well written...

The authors' Answer: Thanks for the kind advice. We have invited a professional organization to help modify the language. The confusing, conflict, and unclear technical details will be stated clearly.

2. Specific comments

The reviewer's comment 1: The abstract was not well written. (1) 'To estimate streamflow without observation, the authors extend existing techniques . . .', but it is not clear what is the existing one and what is extended one? A simple 'coupling hydrological model with a hydrodynamic model' is not far clear. (2) L13-15: It is hard to understand. Wha t is land covered area? (3) L15-17: I still did not get what the original and adjusted scenarios are. (4) it is not that convincing to say R2 with higher values and bias with lower values, it would be better to use numbers or a range (e.g., 0.7âĹij0.8).

The authors' Answer: (1)The sentence may be not clearly written. It has been revised as follows: To solve the problem of estimating and verifying stream flows without direct observation data; we estimating stream flows in ungauged zones by linking a hydrological model with a hydrodynamic model, taking the Poyang Lake basin as a test case. To simulate streamflow of the ungauged zone, we build a SWAT model for the entire catchment area covering the upstream gauged area and the ungauged zone; then to calibrate the SWAT model using the gauged area. To verify the results, we built two hydrodynamic scenarios (the original and adjusted scenarios) for Poyang Lake using Delft3D model. In the original scenario, the upstream boundary condition is the observed streamflow of the upstream gauged area; while it is the summation of the observed streamflow and the simulated ungauged streamflow in the adjusted scenario. (2) Land covered area means the area which is not covered by water body. Seen in Figure 1, the land covered area of the ungauged zone is the area inside the yellow line and outside the boundary of Poyang Lake. Originally, the Poyang Lake ungauged zone includes two parts: the land covered area and the Poyang Lake. As the lake water level and streamflow is model by the lake dynamic model (Delft3D model), we do not need to calculate the streamflow of Poyang Lake and set the streamflow as the input
of the lake dynamic model. So we redefined region of the ungauged zone as the area inside the yellow line and outside the Poyang Lake (Figure 1(a)). The area does not include the Poyang Lake. L13-15 is revised as the follows. To simulate streamflow of the ungauged zone, we build a SWAT model for the entire catchment area covering the upstream gauged area and the ungauged zone; then calibrate the SWAT model using the gauged area. (3) Thank you for the valuable suggestion. There may exist some writing problems here. The method should be described in details. The procedures of the manuscript are as follows. Procedure 1: we first calculated the streamflow produced by the Poyang Lake ungauged zone (PLUZ). Procedure 2: we compared the model result (water level and discharge in outlet (Hukou)) of the dynamic model with and without the ungauged streamflow. For Procedure 1, to simulate the streamflow in the Poyang Lake ungauged zone we build a SWAT model for the entire catchment covering the upstream gauged zone and the ungauged zone. The parameters were calibrated using the observed streamflow from the gauges located in the upper streams. For Procedure 2, the simulated streamflow in PLUZ were used as part of the inflows for hydrodynamic model to simulate the water level and other hydrodynamic characteristics of Poyang Lake. In this processing, two lake hydrodynamic scenarios (the Adjusted Scenario, the Original Scenario) are constructed. In Adjusted Scenario, the upper inflow boundary of hydrodynamic model is the summation of the simulated ungauged streamflow and the upstream gauged streamflow. In Original Scenario, the upper inflow boundary of the hydrodynamic model is the upstream gauged streamflow. The modeled results (water level and discharge of outlet (Hukou)) in Adjusted Scenario, Original Scenario were compared. In summary, the Adjusted Scenario take the ungauged streamflow into consideration while the Original Scenario does not. In Adjusted Scenario, inflows for hydrodynamic model is the summation of simulated streamflow in PLUZ and the observed streamflow from gauges located in the upper streams. In Original Scenario, inflows for hydrodynamic model is the observed streamflow from gauges located in the upper streams. The model in the Original Scenario has been calibrated and validated. The model in the Adjusted Scenario use the same parameter as that in the

Original Scenario. (4)The sentence has been revised as follows: Experimental results show there was a narrower discrepancy (R2=0.81, PBIAS=10.00%) between the stream flows observed at the outlet of the lake and the simulated stream flows in adjusted scenarioïjŇcompared to that in original scenario (R2=0.77, PBIAS=20.10%).

The reviewer's comment 2: L29- 33: it does not read well, the connection seems not logical.

The authors' Answer: The sentences are revised as follows: In order to reduce the damage to the population, agriculture and economy, we should predict floods and droughts precisely. However, in watersheds there is an ungauged zone lacking stream flow observations. The streamflow of ungauged zone is difficult to estimate, which makes ungauged zones neglected in water yield estimation. Therefore, it is important to estimation the streamflow in the ungauged zone.

The reviewer's comment 3: L66-67: What does 'Usually, there are stream flow observation at the lower boundary of the ungauged zone.' Mean?

The authors' Answer: The sentence may not be written clearly. The sentence is revised as follows: Usually, the downstream of the ungauged zone exist a lake (or a river, an ocean). The lake is gauged by streamflow gauging stations at the outlet and water level gauging stations on the water surface.

The reviewer's comment 4: L72-75: Dargahi and Setegn combined a hydrological model (SWAT) with a 3D hydrodynamic model (GEMSS) .... Bellos and Tsakiris . . ... However, . . . there is no clear and specific method of coupling hydrological and hydrodynamic models in space and time. It is really hard for readers to get what problems or drawbacks others have, and what the novelty of the authors' method is.

The authors' Answer: The sentences should be far clear. They has been revised as follows: Dargahi and Setegn combined a hydrological model (SWAT) with a 3D hydrodynamic model (GEMSS) .... Bellos and Tsakiris . . ... However, the method combing

hydrological model and hydrodynamic model is scarcely applied in the ungauged zone for streamflow simulation and validation. As the ungauged zone is usually in flat topography with turbulent flow, it is difficult to draw watersheds in the ungauged zone. What's more, allocating the streamflow in the ungauged zone to inflow boundary of hydrodynamic model is not an easy work. How to drawing watersheds and allocating the streamflow are not mentioned in the previous researches. The detail of linking hydrology and hydrodynamic models in the ungauged are presented in the study.

The reviewer's comment 5: L101-103: 'We established . . . model was established to . . .' Grammar issue.

The authors' Answer: The sentences has been revised as follows: We established two lake hydrodynamic scenarios to further verify the streamflow simulation results.

The reviewer's comment 6: L103-106: It is strange the end of Introduction was repeating the abstract.

The authors' Answer: The related sentences has been deleted.

The reviewer's comment 7: L121-124: It reads awkward, and it seems SWAT doesn't need temperature? Were all the data downloaded from Jiangxi hydro info website?

The authors' Answer: (1) The sentences may be confused. They has been revised as follows: Data required by the SWAT model include the forcing elements of daily rainfall, evapotranspiration, temperature, relative humidity and wind from 1980 to 2014 collected at 16 national meteorological stations. The stations are distributed uniformly across the area (Fig. 1a). This data was downloaded from China Meteorological Data Sharing Service System (http://data.cma.cn/). (2) No. Daily rainfall, evapotranspiration, temperature, relative humidity and wind data were downloaded from China Meteorological Data Sharing Service System. Streamflow data at 7 gauging stations (Qiujin, Wanjiabu, Waizhou, Lijiadu, Meigang, Hushan, and Dufengkeng), daily observation for water level at water surface stations (Xingzi, Duchang and Kangshan), and outflow

discharges at Hukou were downloaded from Jiangxi hydro info website.

The reviewer's comment 8: Methodology section is too short and lack details.

The authors' Answer: The section should be clearer. We will add more details and reorganize the methodology section clearly.

The reviewer's comment 9: SWAT and Delft3D are the two major approaches of the study; however, there was no description of the two models.

The authors' Answer: The descriptions for the two models have been added in the manuscript. The part described for SWAT is as follows: We used SWAT (Soil and Water Assessment Tool) (Arnold et al., 1993) model to simulate stream flows in PLUZ. SWAT was physically-based, semi-distributed and river basin-scale hydrological model. It was developed to assess the impact of land management practices on stream flow, sediment and agricultural yields in complex basins with changing soil type, land use and manage over long time. For purpose of modelling, an entire watershed is divided into subwatersheds based on rivers and DEM data. Subwatershes are portioned into Hydrological Response Units (HRUs), the minimum research units. Water balance is the driving force of hydrological processes. Hydrological cycle including two division: runoff producing on land and flow routing in channel. Surface runoff volume is calculated using SCS method (USDA Soil Conservation Service, 1972). Flow routed through the channel is calculated by variable storage coefficient method (Williams et al., 1969). SWAT has already been applied to watersheds widely in the world for stream flow simulation (Douglas-Mankin et al., 2010;Arnold et al., 2012;Luo et al., 2016). The part described for Delft3D is as follows: Delft3D simulates the hydrodynamic pattern via the Delft3D-FLOW (Roelvink and van Banning, 1994) module. Delft3D-FLOW is a multi-dimensional (two dimension or three dimension) hydrodynamic and transport simulation programme. The programme can calculate unsteady flow by building linear or curvilinear grid suitable for water boundary, which is forced by tidal and meteorological data. Delft3D-FLOW is based on the Reynolds-Averaged Navier-Stokes

(RANS) equations, which is simplified for an incompressible fluid under shallow water and Boussinesq assumptions. The RANS equations are solved by alternative direction implicit finite difference method (ADI) on spherical or orthogonal curvilinear grid. Delft3D has ability to simulate water level variations and flows on surface water bodies in response to forcing elements of inflow discharges and climate factors. It has been proven by application on many surface water bodies around the world. Delft3D is considered appropriate for the wide and shallow characteristics of Poyang Lake.

The reviewer's comment 10: L146-147: to simulated ?

The authors' Answer: As the PLUZ does not include Poyang Lake. The sentence has been revise as follows: We used SWAT model to simulate stream flows in the land covered area of the PLUZ.

The reviewer's comment 11: The results and discussion seems just result description and no discussion was provided.

The authors' Answer: We will revise the discussion part. The application of the method and the influence factors on the simulation results will be discussed, as well as the impact of climate change on the hydrodynamic characteristics on the Poyang Lake.

The reviewer's comment 12: There are many grammar issues here and there, and I believe they need a professional editing service before resubmission.

The authors' Answer: We have invited a professional editing service to revise the grammar issues.

---

## Author Response (AR1)

**Replies to Referee #1**

We are very grateful to the reviewers for reading the manuscript extremely carefully and forwarding the valuable suggestions for improvement. Point-by-point responses to the reviewers' comments are listed below.

5

**1. General comment**

The reviewer's comment 1: ...but the poor writing make it hard to be understood well. ...However, the manuscript was not well written...

10 **The authors' Answer:** Thanks for the kind advice. We have invited a professional organization to help modify the language. The confusing, conflict, and unclear technical details has been stated clearly.

**2. Specific comments**

15 **The reviewer's comment 1**: The abstract was not well written. (1) 'To estimate streamflow without observation, the authors extend existing techniques . . .', but it is not clear what is the existing one and what is extended one? A simple 'coupling hydrological model with a hydrodynamic model' is not far clear. (2) L13-15: It is hard to understand. What is land covered area? (3) L15-17: I still did not get what the original and adjusted scenarios are. (4) it is not that convincing to say R2 with higher values and bias with lower values, it would be better to use numbers or a range (e.g., 0.7~0.8).

**20 The authors' Answer:**

(1)The sentence may be not clearly written. It has been revised as follows:

To solve the problem of estimating and verifying streamflow without direct observation data, we estimated streamflow in ungauged zones by coupling a hydrological model with a hydrodynamic model, using the Poyang Lake Basin as a test case. To simulate the streamflow of the ungauged zone, we built a SWAT model for the entire catchment area covering the upstream

- 25 gauged area and ungauged zone; and then calibrated the SWAT model using the data in the gauged area. To verify the results, we built two hydrodynamic scenarios (the original and adjusted scenarios) for Poyang Lake using the Delft3D model. In the original scenario, the upstream boundary condition is the observed streamflow from the upstream gauged area, while it is the sum of the observed from the gauged area and the simulated from the ungauged zone in the adjusted scenario.
- (2) Land covered area means the area which is not covered by water body. Seen in Figure 1, the land covered area of the ungauged zone is the area inside the yellow line and outside the boundary of Poyang Lake. Originally, the Poyang Lake ungauged zone includes two parts: the land covered area and the Poyang Lake. As streamflow should be considered by the lake dynamic model (Delft3D model), we do not need to calculate the streamflow of Poyang Lake and set the streamflow as the input of the lake dynamic model. So we redefined region of the ungauged zone as the area inside the yellow line and outside the Poyang Lake (Figure 1(a)). The area does not include the Poyang Lake. L13-15 is revised as the follows.
- To simulate the streamflow of the ungauged zone, we built a SWAT model for the entire catchment area covering the upstream gauged area and ungauged zone; and then calibrated the SWAT model using the data in the gauged area.

(3) Thank you for the valuable suggestion. There may exist some writing problems here.

The method should be described in details. The procedures of the manuscript are as follows.

- Procedure 1: we first calculated the streamflow produced by the Poyang Lake ungauged zone (PLUZ). Procedure 2: we compared the model result (water level and discharge in outlet (Hukou)) of the dynamic model with and without the ungauged streamflow. For Procedure 1, to simulate the streamflow in the Poyang Lake ungauged zone we build a SWAT model for the entire catchment covering the upstream gauged zone and the ungauged zone. The parameters were calibrated using the observed streamflow from the gauges located in the upper streams.
- For Procedure 2, the simulated streamflow in PLUZ were used as part of the inflows for hydrodynamic model to simulate the water level and other hydrodynamic characteristics of Poyang Lake. In this processing, two lake hydrodynamic scenarios (Adjusted Scenario, Original Scenario) are constructed. In Adjusted Scenario, the upper inflow boundary of hydrodynamic

model is the summation of the simulated ungauged streamflow and the upstream gauged streamflow. In Original Scenario, the upper inflow boundary of the hydrodynamic model is the upstream gauged streamflow. The modeled results (water level and ouflow) in Adjusted Scenario and Original Scenario were compared.

- 50 In summary, the Adjusted Scenario take the ungauged streamflow into consideration while the Original Scenario does not. In Adjusted Scenario, inflows for hydrodynamic model is the summation of simulated streamflow in PLUZ and the observed streamflow from gauges located in the upper streams. In Original Scenario, inflows for hydrodynamic model is the observed streamflow from gauges located in the upper streams. The model in Original Scenario has been calibrated and validated. The model in Adjusted Scenario use the same parameter as that in Original Scenario.
- 55 (4)The sentence has been deleted. The similar sentence is written as you comment.

The reviewer's comment 2: L29- 33: it does not read well, the connection seems not logical.

The authors' Answer: The sentences are revised as follows:

To reduce the damage to the population, agriculture and economy, we should attempt to predict floods and droughts precisely.
 However, in watersheds, there is ungauged zones lack streamflow observations. The ungauged streamflow is difficult to estimate and is usually neglected in water yield estimations, which can result in floods/droughts predictions being not accurate enough.

**The reviewer's comment 3:** L66-67: What does 'Usually, there are stream flow observation at the lower boundary of the ungauged zone.' Mean?

The authors' Answer: The sentence may not be written clearly. The sentence is revised as follows: Usually, a water body (a lake, a river, or an ocean) exists downstream of the ungauged zone. The water body is gauged by streamflow gauging stations at the outlet and water level gauging stations on the water surface.

70 **The reviewer's comment 4:** L72-75: Dargahi and Setegn combined a hydrological model (SWAT) with a 3D hydrodynamic model (GEMSS) .... Bellos and Tsakiris . . .. However, . . . there is no clear and specific method of coupling hydrological and hydrodynamic models in space and time. It is really hard for readers to get what problems or drawbacks others have, and what the novelty of the authors' method is.

The authors' Answer: The sentences should be clearer. They has been revised as follows:

- 75 Dargahi and Setegn (2011) combined a watershed hydrological (SWAT) model with a 3D hydrodynamic model (GEMSS) to simulate the Tana Lake Basin to address the impact of climate change. Bellos and Tsakiris (2016) combined hydrological and hydrodynamic techniques for flood simulation in the Halandri catchment. However, the method combing a hydrological model and a hydrodynamic model is rarely applied in such ungauged zone. As the ungauged zone is usually located in flat topography with turbulent flow, it is difficult to draw watersheds in the ungauged zone. In addition, allocating the ungauged streamflow to
- 80 the inflow boundary of a hydrodynamic model is not easy. The ways to drawing watersheds and allocating the streamflow are not mentioned in the previous studies. The details of coupling hydrology and hydrodynamic models in the ungauged are presented in the study.
  - The reviewer's comment 5: L101-103: 'We established . . . model was established to . . .' Grammar issue.

85 The authors' Answer: The sentences has been revised as follows:We established two lake hydrodynamic scenarios to further verify the streamflow simulation results.

**The reviewer's comment 6:** L103-106: It is strange the end of Introduction was repeating the abstract. **The authors' Answer:** The related sentences has been deleted.

90

65

The reviewer's comment 7: L121-124: It reads awkward, and it seems SWAT doesn't need temperature? Were all the data downloaded from Jiangxi hydro info website?

The authors' Answer:

(1) The sentences may be confused. They has been revised as follows:

- 95 Data required by the SWAT model include the forcing elements of daily rainfall, evapotranspiration, temperature, relative humidity and wind from 1980 to 2014 collected at 16 national meteorological stations. The stations are distributed uniformly across the area (Fig. 1a). This data was downloaded from China Meteorological Data Sharing Service System (http://data.cma.cn/).
- (2) No. Daily rainfall, evapotranspiration, temperature, relative humidity and wind data were downloaded from China
   Meteorological Data Sharing Service System. Streamflow data at 7 gauging stations (Qiujin, Wanjiabu, Waizhou, Lijiadu, Meigang, Hushan, and Dufengkeng), daily observation for water level at water surface stations (Xingzi, Duchang and Kangshan), and outflow discharges at Hukou were downloaded from Jiangxi hydro info website.

The reviewer's comment 8: Methodology section is too short and lack details.

105 **The authors' Answer:** The section should be clearer. We have added more details and reorganize the methodology section clearly.

The reviewer's comment 9: SWAT and Delft3D are the two major approaches of the study; however, there was no description of the two models.

110 The authors' Answer: The descriptions for the two models have been added in the manuscript.

The part described for SWAT is as follows:

We used a SWAT (Soil and Water Assessment Tool) (Arnold et al., 1993) model to simulate streamflow in the PLUZ. SWAT is a physically-based, semi-distributed and river basin-scale hydrological model. It is developed to assess the impact of land management practices on streamflow, sediment and agricultural yields in complex basins with changing soil types, land use

- and management over long periods of time. For the purpose of modelling, an entire watershed is divided into sub-watersheds based on rivers and DEM data. Sub-watersheds are portioned into Hydrological Response Units (HRUs), the minimum research units. Water balance is the driving force of hydrological processes. The hydrological cycle includes two divisions: runoff-producing on land and flow-routing in channels. The surface runoff volume is calculated using the SCS method (USDA Soil Conservation Service, 1972). Flow routed through the channel is calculated by the variable storage coefficient method
- 120 (Williams et al., 1969). SWAT has already been widely applied to watersheds around the world for streamflow simulation (Douglas-Mankin et al., 2010; Arnold et al., 2012; Luo et al., 2016).

The part described for Delft3D is as follows:

Delft3D simulates the hydrodynamic pattern via the Delft3D-FLOW (Roelvink and van Banning, 1994) module. Delft3D-FLOW is a multi-dimensional (two-dimension or three-dimension) hydrodynamic and transport simulation program. The

- 125 program can calculate unsteady flow by building linear or curvilinear grids suitable for the water boundary, which is forced by tidal and meteorological data. Delft3D-FLOW is based on the Reynolds-Averaged Navier-Stokes (RANS) equations, which are simplified for an incompressible fluid under shallow water and Boussinesq assumptions. The RANS equations are solved by the alternative direction implicit finite difference method (ADI) on a spherical or orthogonal curvilinear grid. Delft3D has ability to simulate water-level variations and flows on surface water bodies in response to forcing elements of inflow discharges
- 130 and climate factors, which has been proven by applications on many surface water bodies around the world. Delft3D is considered appropriate for the wide and shallow characteristics of Poyang Lake.

**The reviewer's comment 10: L146-147: to simulated ?**

The authors' Answer: As the PLUZ does not include Poyang Lake. The sentence has been revise as follows:

135 We used SWAT model to simulate stream flows in the land covered area of the PLUZ.

The reviewer's comment 11: The results and discussion seems just result description and no discussion was provided. The authors' Answer: We have revise the discussion part. The intra-annual and inter-annual variation of the ungauged streamflow will be discussed, as well as the impact to the lake water balance.

The reviewer's comment 12: There are many grammar issues here and there, and I believe they need a professional editing service before resubmission.

The authors' Answer: We have invited a professional editing service to revise the grammar issues.

We are very grateful to the reviewer for reading the manuscript extremely carefully and forwarding the valuable suggestions for improvement. Point-by-point responses to the reviewers' comments are listed below.

**1. General Comments**

150

160

165

The reviewer's comment 1: However, the writing is very poor. Most of sentences are too awkward to be understood even though the grammar of the sentence is correct. After reading the whole of methodology, I could not proceed with the rest of the manuscript, not only because of the poor English writing but also because of the confusing, conflict, and unclear technical details.

155 **The authors' Answer:** Thanks for the kind advice. We have invited a professional organization to modify the language.

The reviewer's comment 2: Based on my understanding, the methods in this study should be very clear: authors firstly simulated the streamflow of inlets for Poyang Lake using SWAT model; the parameters were calibrated using the observed streamflow from gauges located in the upper streams; then, the simulated streamflow were used as the inflows for hydrodynamic model to simulate the water level and other hydrodynamic characteristics of Poyang Lake; finally, modeled

water level and discharge in outlet (Hukou) with and without SWAT simulated inflows were compared. If my understanding is correct...

**The authors' Answer:** The reviewer is almost right. The procedures of the manuscript are as follows. Procedure 1: we first calculated the streamflow produced by the Poyang Lake ungauged zone (PLUZ). Procedure 2: the modeled water level and discharge in outlet (Hukou) with and without PLUZ streamflow were compared.

For Procedure 1, the Poyang Lake ungauged zone including two parts. One is the land cover area of PLUZ, the other one is water covered area (It has been revised as Poyang Lake). The water covered area is Poyang Lake. So in the next step, we calculate the streamflow for the land covered area and Poyang lake separately.

The streamflow for the land covered are of PLUZ is simulated by SWAT model. The parameters were calibrated using the

170 observed streamflow from gauges located in the upper streams. The streamflow in the land coved area of PLUZ is calculated as the difference value of simulated streamflow at inlets of Poyang Lake and the observed streamflow of the upstream gauged area.

The streamflow for the Poyang Lake is calculated by a simplified equation (Eq (1)). As the reviewer comments, the equation is not serious for not considering the lake storage change (This will be discussed in **the reviewer's comment 4 in General**

175 **Comments**).

The steamflow of PLUZ is the summation of streamflow produced by the land coved area and Poyang Lake.

For Procedure 2, the calculated streamflow in PLUZ were used as part of the inflows for hydrodynamic model to simulate the water level and other hydrodynamic characteristics of Poyang Lake. In this processing, two lake hydrodynamic scenarios (Adjusted Scenario, Original Scenario) are built. In Adjusted Scenario, inflows for hydrodynamic model is the summation of

180 simulated streamflow in PLUZ and the observed streamflow from gauges located in the upper streams. In Original Scenario, inflows for hydrodynamic model is the observed streamflow from gauges located in the upper streams. The modeled results (water level and discharge) in Adjusted Scenario and Original Scenario were compared.

The details is in section 3 in the manuscript.

185 **The reviewer's comment 3:** ...I don't understand why authors used more than five pages to describe this simple procedure and the procedure hasn't been clarified ultimately. For example, I didn't see any coupling of hydrological and hydrodynamic models in sections 3.3

**The authors' Answer:** Thank you for the valuable suggestion. The coupling is in L197-252. The writing of manuscript may confuse you.

In the manuscript, it is loosen coupling. In the Adjusted Scenario, the output of SWAT model is the input of Delft3D model.
The streamflow of PLUZ is used as part of the upper inflows for hydrodynamic model.
In the Original Scenario when the streamflow of PLUZ is not considered, there are 9 inflow points (*d1*, *d2*, *d3*, *d4*, *d5*, *d6*, *d7*, *d8 and d9*) for the lake (Figure 1(b)). Inflows to the points comes from the 7 gauging stations (Qiujin, Wanjiabu, Waizhou, Lijiadu, Meigang, Hushan and Dufengkeng). In the Adjusted Scenario, we should solve the problem: how to allocate the

195 ungauged streamflow in to different inflow points.

As the ungauged zone is usually in flat topography with turbulent flow, it is difficult to draw watersheds in the ungauged zone. What's more, allocating the streamflow in the ungauged zone to inflow boundary of hydrodynamic model is not an easy work. The coupling sections (section 3.4) describe the content: how to drawing watersheds in the ungauged zone and allocating the ungauged streamflow to the lake model properly.

200 The section may confuse you. We have reorganized the coupling section.

**The reviewer's comment 4:** ...but a lot of confusing water balance equations (i.e., Equation (2-4)), especially for equation (3). Where is the water level change (i.e., surface water storage) in equation (3)? If the water level change is too small to be

negligible authors may need to verify it and clarify it in the manuscript. In my opinion, all the Equation (2-4) can be written in

one if considering the Poyang Lake as the control volume: QHukou = P+Qinflows-E-ΔSWSsimulated where QHukou is the discharge in the outlet of Poyang lake; P is the precipitating in Poyang Lake; Qinflows is the summation of the streamflow in all inlets of Poyang Lake; E is the evaporation in Poyang Lake; ΔSWSsimulated is the water level changes in Poyang Lake.
 The authors' Answer: Thank you very for the valuable suggestion. The writing of manuscript may confuse you.

In Section 3.3 (Equation (2-4)), we intended to calculated the inflow at different inflow points of the lake in the Adjusted Scenario. The inflow at the point was the summation of three parts: the streamflow in land covered PLUZ, the streamflow in Poyang Lake, and the streamflow from the upper gauged stream (the 7 gauging stations: Qiujin, Wanjiabu, Waizhou, Lijiadu, Meigang, Dufengkeng and Hushan).

However, originally I was confused by the concept of STREAMFLOW in Poyang Lake. The streamflow in Poyang Lake should take the lake level change into consideration. However, the hydrodynamic model modeled the changed lake level and outflow in long time series. The streamflow of the lake should not be the upper inflow boundary of the Lake hydrodynamic

215 outflow in long time series. The streamflow of the lake should not be the upper inflow boundary of the Lake hydrodynamic model, it should be considered by the hydrodynamic model. But the hydrodynamic model is developed without considering evaporation. And we estimated the evaporation on the surface of the lake as less than 2% of the total water resource. Thus, in our study, we have ignored the evaporation on the surface of the lake.

So we revise the inflow of the lake as the summation of two parts: the streamflow in land covered area of PLUZ, and the streamflow from the upstream gauged area (the 7 gauging stations).

220

As the lake hydrodynamic model can model the changed lake level and outflow, we have decided not to take Poyang Lake as part of the ungauged zone. So the PLUZ is redefined the area inside the yellow line and outside the boundary of Poyang Lake (Figure 1(a)).

- 225 The reviewer's comment 5: ...Given the authors' methodology is correct, I have another two concerns: has the hydrodynamic model been calibrated? If yes, the bias and error of the simulated ungauged streamflow can be corrected during the calibration of hydrodynamic model which means the verification of simulated streamflow in ungagged zone may be spurious; The authors' Answer: Thank you very for the valuable suggestion. The writing of manuscript may make you confused. We construct two scenarios (Adjusted Scenario, Original Scenario) for the lake hydrodynamic model. Adjusted Scenario take
- 230 the ungauged streamflow into consideration while Original Scenario does not. The model in Original Scenario has been calibrated and validated. The model in Adjusted Scenario use the same parameter as that in Original Scenario.

**The reviewer's comment 6:** the other concern is I don't think it's necessary to use SWAT simulation as the results shown in Figure 4 since the discrepancy of two scenarios with and without SWAT is relatively small which may be smaller than the

235 uncertainties in SWAT simulations as shown in Figure 3.

The authors' Answer: Thank you for the comments. The writing may confuse you. If the writing is more clear, your comments is as follows: the other concern is I don't think it's necessary to use SWAT simulation as the results shown in Figure 4 since the discrepancy of two scenarios with and without ungauged streamflow is relatively small which may be smaller than the uncertainties in SWAT simulations as shown in Figure 3.

- The SWAT and Delft3D are two different models and applied in different specific fields. The acceptable simulation accuracies are different too. In general, SWAT model simulation can be judged as satisfactory if Ens (Nash-Sutcliffe efficiency) > 0.50 and absolute PBIAS (percent bias) < 25% for streamflow (Van Liew et al. 2007), while for water level (or tide level) simulation by Delf3D model Ens can be larger than 0.90 (some can reach 0.99) and absolute PBIAS can be smaller than 5% (Qi et al., 2016; Zhang et al., 2015). The Delf3D model is steady and the uncertainty is relatively small. Therefore, it is not easy to
- 245 improve the simulation result of Delft3D model. If the simulated accuracy of Delft3D model can be increased, the increased range should be small and may be smaller than the uncertainty of SWAT model.

However, it is valuable to improve the Delft3D model, although the increased range is small. In the previous research for Poyang Lake dynamic model, the simulated streamflow at the outlet (Hukou) is smaller than the observed in average (Qi et al., 2016; Zhang et al., 2015). This may result from not taking the ungauged streamflow into consideration. So it is valuable and important to test if ungauged streamflow can increase the Poyang Lake dynamic model. And we are curious about what the

250 important to test if ungauged streamflow can increase the Poyang Lake dynamic model. And we are curious about what the results would be. It is important to get the result: the ungauged zone improves the Delft3D model accuracy although the increase range of the accuracy is small.

You comment means the discrepancy should be bigger than the SWAT uncertainty if we want to validate the simulation result of the ungauged streamflow by Delft3D model. That is because the increased discrepancy may be caused by the uncertainty

of the SWAT model?

If my understanding is right, what we should do is to demonstrate that the discrepancy (the increased accuracy) is not the random fluctuation result of the SWAT uncertainty. We statistic the annual  $R^2$  (determination coefficient) and monthly RMSE (root mean square error) in section 4.2. Mostly  $R^2$  is bigger and RMSE is smaller in Adjusted Scenario than that in Original Scenario. Furthermore, in section 4.4 the ungauged streamflow did improve the lake water balance. The evidences show the improvement in Adjusted Scenario is not random. It suggested the ungauged streamflow is reasonable.

260

**2. Specific Comments**

The reviewer's comment 1: Line 15: Is the water covered area of the ungauged zone the Poyang Lake? If yes, please revise

265 or it is very confusing.

The authors' Answer: Yes. It will been revised in the manuscript.

The reviewer's comment 2: Line 18: how do you conclude "narrower discrepancy"? Please provide some quantification. The same for Line 23 "higher value"

270 The authors' Answer: Yes. It has been revised in the manuscript.

The reviewer's comment 3: Line 29-30: Please rewrite this sentence.

The authors' Answer: The sentence is revised in the manuscript.

To reduce the damage to the population, agriculture and economy, we should attempt to predict floods and droughts precisely.

275 However, in watersheds, there is ungauged zones lack streamflow observations. The ungauged streamflow is difficult to estimate and is usually neglected in water yield estimations, which can result in floods/droughts predictions being not accurate enough.

The reviewer's comment 4: Line 40: Please delete the second "stream flow".

280 **The authors' Answer:** It has been delete.

The reviewer's comment 5: Line 51-52: Please revise this sentence. Awkward.

The authors' Answer: Some researchers use regionalization methods to simulate streamflow in ungauged zones. The parameters in the gauged areas are calibrated. Then, the parameters are transformed from gauged to ungauged areas.

285

The reviewer's comment 6: Line 60: Where is the citation for Ma's study?

The authors' Answer: It is an article in Chinese. The reference is as follows:

Ma, X., Liu, D.: Modeling of interval runoff in the region of Dongting Lake[J]. Journal of Hydroelectric Engineering, 30(5):10-15, 2011.

**290**

**The reviewer's comment 7:** Line 124: Please provide the spatial resolution for DEM. **The authors' Answer:** The spatial resolution for DEM is 90 m.

The reviewer's comment 8: Line 131: Please provide some examples about the topographic data.

- 295 The authors' Answer: The following examples of topographic data have been added. The reference is as follows: Qi, H., Lu, J., Chen, X., et al. Water age prediction and its potential impacts on water quality using a hydrodynamic model for Poyang Lake, China. Environmental Science and Pollution Research, doi:10.1007/s11356-016-6516-5, 23(13):13327-13341, 2016.
- Zhang, P., Lu, J., Feng L., et al. Hydrodynamic and inundation modeling of China's largest freshwater lake aided by remote sensing data. Remote Sensing, doi:10.3390/rs70404858, 7(4): 4858-4879, 2015.

The reviewer's comment 9: Line 134-135: Please provide the temporal scales for water level and discharge.
The authors' Answer: Daily scale. The sentence has been revised as follows:
The daily observation for water level (at stations of Xingzi, Duchang and Kangshan), and outflow discharges (at Hukou) from 2000 to 2011 were got from Web of hydrological information in Jiangxi.

The reviewer's comment 10: Line 142: What does "sing" mean here?

The authors' Answer: It is a spelling mistake. It should be "using".

310 The reviewer's comment 11: Line 163: Where is the water level change (i.e., surface water storage change) in equation (1) and (3)? If the water level change can be negligible please verify it and clarify it. (???)
The authors' Answer: The lake water level change has been taken into consideration in the lake hydrodynamic model. As there are observed stations gauging the water level and lake hydrodynamic model can model the streamflow in Poyang Lake, Poyang Lake is not considered as ungauged zone. (more details in Comment 4 in General Comments)

315

305

The reviewer's comment 12: I didn't read the Result section word by word please carefully read it and revise it based on the revised methodology.

The authors' Answer: Thank you for the suggestion. We have modified it based on the the methodology.

320 The reviewer's comment 13: Line 440: Please delete Table 3.

The authors' Answer: We have deleted Table 3 in the revised version.

The reviewer's comment 14: Line 444: Please switch the Figure 1a and 1b; put figure 1a in left hand side; delete the "Meteorological stations" in legend of Figure 1b since there is no meteorological station; add the scale bar in Figure 1b.

325 The authors' Answer: There is a meteorological station of Boyang. And the rest have been revised in Figure 1.

The reviewer's comment 15: Line 450: Figure 2 is confusing, and please revise it. (???) The authors' Answer: It has been revise as the comment (Figure 2).

- 330 The reviewer's comment 16: Please provide the line number for figure 3 and also the captions for the subfigures. The authors' Answer: We will provide the line number for Figure 3. The added captions is as follows: Subfigures (a),(b),(c),(d),(e) and (f) are the calibration and validation result for stations at Wanjiabu, Waizhou, Lijiadu, Meigang, Hushan, and Dufengkeng separately.
- 335 The reviewer's comment 17: Based on Figure 4, I don't think it's necessary to use SWAT simulation. The discrepancy of two scenarios is relatively small which may be smaller than the uncertainties in SWAT model as shown in Figure 3.
  The authors' Answer: We have made more analysis (more details in the reviewer's comment 6 in General Comments).

**The reviewer's comment 18:** Please delete the figures 5-7.

340 **The authors' Answer:** We have deleted the figures 6-7.

**Stream flow simulation and verification in ungauged zones by coupling hydrological and hydrodynamic models: a case study of the Poyang Lake ungauged zone**

Ling Zhang1, Jianzhong Lu1,\*, Xiaoling Chen1,2, Sabine Sauvage3, Jos é-Miguel Sanchez Perez3

345 1State Key Laboratory of Information Engineering in Surveying, Mapping and Remote Sensing, Wuhan University, Wuhan 430079, China

2Key Laboratory of Poyang Lake Wetland and Watershed Research, Ministry of Education, Jiangxi Normal University, Nanchang 330022, China

3ECOLAB, Universit éde Toulouse, CNRS, INPT, UPS, 31400 Toulouse, France

350 \* *Correspondence to*: Jianzhong Lu (lujzhong@whu.edu.cn)

Abstract. To solve the problem of estimating and verifying stream flowsstreamflow without direct observation data; we extend existing techniques for estimating stream flowsestimated streamflow in ungauged zones, by coupling a hydrological model with a hydrodynamic model, using the Poyang Lake basinBasin as a test case. We simulated stream flows in the land covered areaTo simulate the streamflow of the ungauged zone by building, we built a SWAT model for the entire catchment 355 area covering the upstream gauged stations area and the land covered area; then estimated stream flows in the water covered area of the ungauged zone; and then calibrated the SWAT model using the simplified water balance equation data in the gauged area. To verify the results, we built two hydrodynamic scenarios (the original and adjusted scenarios) for Poyang Lake using the Delft3D model. In this study, the original scenario did not take stream flows in the, the upstream boundary condition is the observed streamflow from the upstream gauged area, while it is the sum of the observed from the gauged area and the 360 simulated from the ungauged zone into consideration, unlike in the adjusted scenario that accounts for the ungauged zones. Experimental. The experimental results showshowed that there wasare a narrower discrepancystronger correlation and lower bias ( $R^2 = 0.81$ , PBIAS = 10.00%) between the stream flows observed at the outlet of the lake and the simulated stream flowsstreamflow in the adjusted scenario, compared to that ( $R^2 = 0.77$ , PBIAS = 20.10%) in the original scenario, suggesting the simulated streamflow of the ungauged zone is reasonable. Using our technique this method, we estimated that the ungauged 365 zonestreamflow of the Poyang Lake produces stream flows of approximately 180 ungauged zone as  $16.4 \pm 6.2$  billion  $m^{3}$ ;/a, representing about 11.4%~11.24% of the annual total water yield of the total inflow from the entire watershed. We also analysed the impact of the stream flowsOf the annual water yield, 70% (11.48 billion m3/a) concentrates in ungauged zone on the wet season, while 30% (4.92 billion m3/a) comes from the dry season. The ungauged streamflow significantly improves the water balance between inflow and outflow with the closing error decreased by 13.48 billion  $m^3/a$  (10.10% of the total annual 370 water resource) from  $30.20 \pm 9.1$  billion m3/a (20.10%) of the lake. These results, incorporating the estimated stream flow in ungauged zone, significantly improved the water balance as indicated by R2 with higher value and percent bias with lower

value, as compared to the results when the stream flows in the ungauged zone were not taken into account,  $R^2$  with lower value and percent bias with higher value.total annual water resource) to 16.72 ±8.53 billion m3/a (10.00% of the total annual water resource). The method can be extended to other lake, river, or ocean basins where observation data is unavailable.

**375 **1 Introduction**

380

In recent years, floods and droughts have occurred frequently (Cai et al., 2015; Tanoue et al., 2016), threatening lives and health, reducing crop <del>yieldyields</del> and hindering economic development (Lesk et al., 2016; Smith et al., 2014). If we know the water yield of watersheds, we can predict and prevent droughts and floods. Therefore, it is necessary to fully understand the water yield of watersheds, in order to To reduce the damage of floods and droughts-to the population, agriculture and economy7, we should attempt to predict floods and droughts precisely. However, in watersheds, there is <del>an</del>-ungauged zone lacking stream flowzones lack streamflow observations. Hydrological model is used The ungauged streamflow is difficult 
[revised manuscript text omitted]
 inof the lake, which makes the stream flowstreamflow peak clipped and time-lagged. This Traditional method is too coarse for stream flow simulation in the PLUZ.

450

435

Attempts has More attempts have been made for accurate and precise stream flow streamflow simulation results in the PLUZ. Huang et al. (2011) developed a runoff-fluexflux model especially for the plain area of the PLUZ. The simulation results were verified by comparing the outflow observation observed streamflow at Hukou with the summationsum of the simulated streamflow in the PLUZ and the measured gauged streamflow of the gauged upstream, on the yearly in an annual scale. The 455 time scale of the verification was coarse;. Furthermore, the water storage and flood regulation function of the lake were not taken into consideration. Guo et al. (2011) simulated the daily runoff of the PLUZ byusing the Variable Infiltration Capacity (VIC) and multiple-input single-out-putoutput system (MSIOMISO) models. The verification was performed by comparing the simulated results with the estimated results. However, the estimated result was derived from the time-lag equation, so it could not replace the observed value exactly, for the following two reasons. The: (1) the time-lag equation was a simple 460 hydrodynamic model for the lake, which is not very accurate. In: (2) in the equation, the streamflow at Hukou was adjusted by a modified coefficient at the annual scale, which is not reasonable to be applied inapply at the daily scale. Most recently, Li et al. (2014) combined the hydrological model (WATLAC) and hydrodynamic model (MIKE), where the streamflow in the ungauged area, was roughly calculated by the runoff coefficient method. However, the ungauged area did not take the water 465 studies that include effective verification of stream flow for streamflow simulations in the PLUZ. In the study, the method of combining hydrological and hydrodynamic models is introduced to solve the simulation results for the PLUZ and verification problem in the PLUZ. Our specific objectives are to: (1) simulate and the verify the streamflow in the PLUZ; (2) analyse the inter-annual and intra annual variations of the ungauged streamflow; (3) analyse the impact of the ungauged streamflow on the lake water balance.

470 The object of this study was to solve the verification problem in stream flow simulation in the PLUZ by combining hydrological and hydrodynamic models. The stream flow simulation of the land covered area in the ungauged zone was conducted by building a SWAT model for the whole catchment covering the gauging stations and the land covered area; while the stream flow in the water covered area of the PLUZ was calculated by a simplified water balance equation. We established two lake hydrodynamic model (Delft3D) to further verify the streamflow simulation results. The hydrological and hydrodynamic model (Delft3D) to further verify the streamflow simulation results. The hydrological and hydrodynamic models were coupled in both space and time. We estimated that the ungauged zone of Poyang Lake produces stream flows of approximately 180 billion m3; representing about 11.4% of the total inflow from the entire watershed. The impacts of stream flows in the PLUZ on the water balance of the catchment-lake system were analysed; and the importance of ungauged zones in hydrological prediction for the whole watershed were verified.

**2 Study area and data**

**480 **2.1 Study area**

Poyang Lake is the largest freshwater lake in  $China_{\tau}$  and is connected with the Yangtze River in the north of Jiangxi provinceProvince. The catchment is covered by the five major river sub-catchments and the ungauged zone-shown, in Fig.  $1_{\tau}(a)$ .

As shown in Fig. 1a, stream flow produced by1(a), the Poyang Lake basin includes three parts: the gauged area (the five major river catchments-are), ungauged zone (the PLUZ) and Poyang Lake. The streamflow of the gauged area was measured by the seven stream flowstreamflow stations-(Qiujin, Wanjiabu, Waizhou, Lijiadu, Meigang, Hushan and Dufengkeng). The PLUZ is a plain area and stretches from the stream flow gaugingseven streamflow stations to the outletboundary of the lakePoyang Lake. The PLUZ covers an area of 19,867 km2, 
[revised manuscript text omitted]

$$\underline{\mathbf{Q}_{in}} + \mathbf{P} - \mathbf{E} + \mathbf{G} + \underline{\bigtriangleup \mathbf{S}} + \mathbf{E}' = \underline{\mathbf{Q}_{out}}$$

where, Qin denotes the inflow from the river basins, P is the precipitation in the lake, △S is the storage change of the lake,
and Qout represents the observed outflow at Hukou of the lake. E' represents the uncertainties in the water balance, which arise from errors in observed data and other components, such as the ungauged streamflow and model uncertainty. E represents the evapotranspiration of the lake, less than 2% of the lake outflow. The E data are obtained from Nachang climatology station. G represents the ground water exchange, only 1.3% of the total water balance (Zhang et al. 2014). Thus, we combine the E, G, and E' as the closing error E. As the summation of Qin, P, and △S can be simulated by the hydrodynamic model, the summation is set as the simulated streamflow at Hukou. Traditionally (in Original Scenario), the Qin omits the ungauged streamflow. The water balance equation can be describe as follows.

 $Q_{SimOut,org} + E_{org} = Q_{out}$

(2)

(1)

where QSimOut,org represents the simulated streamflow at Hukou from the hydrodynamic model in Original Scenario. Eorg represent the uncertainty of the equation, which arising from the ungauged streamflow, E, G, the error in the observe data, and uncertainty of the hydrodynamic model. As the ungauged zone occupies 12% of the total water balance (Li et al. 2014), much larger than the other components (E and G, less than 3.3%), the closing error should be large than zero if the observe data and hydrodynamic model are sufficient accuracy. When the ungauged streamflow is taken account (in Adjusted Scenario), the Oin contains the gauged and the ungauged

streamflow. The water balance equation can be describe as follows.

710  $\underline{Q}_{SimOut,adj} + \underline{E}_{adj} = \underline{Q}_{out}$

705

(3)

where  $Q_{SimOut,adj}$  represents the simulated streamflow at Hukou from the hydrodynamic model in Adjusted Scenario.  $E_{adj}$  represent the uncertainty, which doesn't include the ungauged streamflow. Thus, the absolute value of  $E_{adj}$  should be smaller than that of  $E_{org}$ , if the observe data and hydrodynamic model are sufficient accuracy.

**4 Results and discussion**

**715 4. 1 Calibration and validation of SWAT model and Delft3D model**

In order toTo adjust the models to be applied in the Poyang Lake Basinbasin availably, we undertakeundertook calibration and validation for the SWAT model and the Delft3D model. Table 13 and Fig. 3 shows4 show the calibration and validation resultresults for the SWAT model. The observations and simulations at the six gauging stations (Wanjiabu, Waizhou, Lijiadu, Meigang, Hushan and Dufengkeng,) comes) come to a satisfactory agreement, with an R2 or Ens larger than 0.70 and thean absolute value of PBIAS less than 20%, except for Wanjiabu Station. The agreement are fourthly is also supported by the highly consistence high consistency between the observationobservations and the\_simulation; in terms of amplitude and phase, although the simulated peak streamflow wasdid not accurately matchedmatch the observations, producing underestimation and overestimation (Fig. 34). Nevertheless, the calibration and validation result demonstrates that the SWAT model is generally capable of simulating the streamflow of the catchment.

Table 24 and Fig. 4 shows5 show the calibration and validation resultresults for the Delft3D model. The observations and simulations at the four gauging stations (Xingzi, Duchang, Kangshan, and Hukou) comescome to a satisfactory agreement, with an R2 or Ens larger than 0.70 and thean absolute value of PBIAS less than 25%. The agreement are fourthly is also supported by the highly consistence high consistency between the observation and simulation, although there is an are obvious discrepancy discrepancies during the low water level period (Fig. 4a5a, Fig. 4b5b, Fig. 4e5c) and highthe highly changed flow velocity period (Fig. 4d). This outcome 5d). The mismatch probably arises from the decreased elevation of lake bed from the south to the north and the dynamic variation between wetlands and lake areas. The dynamic variation makescauses the lake to be a river in dry periodperiods and turned to beturn into a lake in flood periodperiods, which is difficulty difficult to be a river in dry period period show turned to beturn into a lake in flood periodperiods.

[revised manuscript text omitted]
 analyseanalyze the impact of the PLUZungauged streamflow on the lake water balance of (seen in section 3.4), we 800 calculate the lake catchment system, we compareclosing errors based on the consistence of the inflow (or the simulated outflow)equation 2 and outflow in two cases. In one case, the inflow (or the simulated outflow) incorporated the streamflow produced by the PLUZ; in the other case, the inflow neglected the 3: Eadj when the ungauged streamflow produced by the PLUZ. Fig. 6, Fig. 7 and Fig. 8 shows the comparison in yearly, monthly and daily scales, respectively.
- In Fig. 6, PBIAS between the Observed and the Estimated-is 19.13%; PBISA between the Observed and the Adjusted Estimation1 is 7.94%. The discrepancy between the Observed and the Estimated is narrower than that between the Observed and the Adjusted Estimation1. The Estimated represent the total streamflow of the seven gauging stations, and the Adjusted Estimation1 represent the summation of streamflow in the PLUZ and total streamflow of the seven gauging stations. PBIAS is decreased and the discrepancy is narrowed, when streamflow in the PLUZ neglected. The result suggests the streamflow in the PLUZ improves the water balance of inflow and outflow of the lake, in yearly scale.
- 810 In Fig. 7, PBIAS between the Observed and the Estimated is 19.13% while PBISA between the Observed and the Adjusted Estimation1 is 7.94%; the discrepancy between the Observed and the Estimated is narrower than that between the Observed and the Adjusted Estimation1. PBIAS is decreased and the discrepancy is narrowed, when streamflow in the PLUZ neglected. The result suggests the streamflow in the PLUZ improves the water balance of inflow and outflow of the lake, in monthly scale.
- 815 However, in monthly scale R2 is decreased from 0.74 when streamflow in the PLUZ is neglected to 0.72 when streamflow in the PLUZ is taken into account. That seem to get a worse relationship between the inflow and the outflow when the PLUZ is taken into account. The result arise from the water storage and flood regulation function of the Poyang Lake in daily scale. So we built hydrodynamic model for the lake, considering the lake function of water storage and flood regulation. The result was shown in Fig. 8.
- 820 In Fig.8, PBIAS between the Observed and the Estimated is 19.13% while PBISA between the Observed and the Adjusted Estimation2 is 7.94%; R2 between the Observed and the Estimated is 19.13% while R2 between the Observed and the Adjusted Estimation2 is 7.94%; the discrepancy between the Observed and the Estimated is narrower than that between the Observed

and the Adjusted Estimation2 in most period. The Adjusted Estimation2 represent the prediction result from the hydrodynamic model in considered (Adjusted Scenario. The PBIAS is decreased, R2 is increased) and the discrepancy is narrowed£org when the ungauged streamflow is omitted (Original Scenario), in the PLUZ was considered. AndFig. 9. As shown in Fig. 9, for Adjusted Estimation2 whenmost months (nearly 83%), the absolute value of  $\mathcal{E}_{adj}$  is smaller than that of  $\mathcal{E}_{org}$ , which can demonstrate the ungauged streamflow in the PLUZ is considered, the blocking effects of Yangtze River are reproduced reasonably. In summary, the streamflow in the PLUZ improve the improves the lake water balance-of the lake obviously...

However, there are some exceptional dot pairs colored in red (outlier, only 17%) in Fig. 9. For the exceptional, the absolute

- 830 Eadj is not less than the absolute Eorg as the above. All the exceptional almost concentrates in the high flow period from July to October (Fig. 9). That is an unstable stage when backward flow from Yangtze River usually appears and the water level of Yangtze River usually keeps high (David et al. 2006), which can result in high dynamical changed flow. Thus, more uncertainties would be added to the measured data and the hydrodynamic model during unstable season (July to October) compared to the stable season (January to June, December to November). High dynamic changed flow may cause the
- streamflow overestimated randomly. High water level of Yangtze River also can leads to overestimated streamflow at Hukou, compared to the conditions in normal water level. What's more, frequent water abstraction for irrigation from July to October can also strength the overestimation situation. The accumulative estimation can even lead the closing error less than zero between July and October (Fig. 9), which is opposite to that the closing error should be more than zero described in section 3.4. The evidence suggests that the hydrodynamic model is not accuracy enough to simulate the streamflow during the unstable season. During the time, the added input component could make the ever overestimated streamflow larger. Thus, the closing error will be extended. That's why when £org is less than zero, the £adj will be more less than zero (the red dot pairs in Fig. 9). The evidence just demonstrates that the hydrodynamic model is not accuracy enough to simulate the lake input components during the unstable season from July to October. It doesn't deny the role of ungauged simulated streamflow in improving the lake water balance.
- The ungauged streamflow decreases the annual average closing error of water balance by 13.48 billion  $m^3/a$  (10.10% of the total annual water resource) from 30.20  $\pm$  9.1 billion  $m^3/a$  (20.10% of the total annual water resource) to 16.72  $\pm$  8.53 billion  $m^3/a$  (10.00% of the total annual water resource) for 2001-2010. The evidence also suggests the ungauged simulated streamflow is reasonable.

**5** Conclusions**

850 MethodA method coupling hydrology and hydrodynamichydrodynamics can be used to simulate and verify stream flowsstreamflow in ungauged zones, solving the simulation and verification problemproblems caused by no-the unavailability of streamflow observations. Ungauged zones lacks stream flow observations for calibration and verification for stream flow simulation. The couple hydrological models for the water body of ungauged zones, can verify the stream flow simulation result of ungauged zones using stream flow observations at the lower boundary of the water body. Due to the verification, the method can demonstrate the reliable of stream flow simulation result of ungauged zone. In the study, discrepancy between the observed and the simulated stream flows of the hydrodynamic model when the ungauged zones was taken into consideration, is narrower than that when the ungauged zones was ignored. The result suggests that the stream flow simulation of the ungauged zone is reliable, verifying the simulation result furtherly.-

The hydrological and hydrodynamic models are coupled seamless seamlessly in both space and time. The method of coupling the models in detail was presented in detail for the first try. Sub basins in the ungauged zones and the gauged zone must be coupled in space. Inflow to the water body is sum of stream flow from the gauged and ungauged zone in daily scale. The method istime and was applied in the case study successfully. Using thethis method, we estimated that the ungauged zone of Poyang Lake produces stream flows a streamflow of approximately 18016.4 billion m35, representing about approximately 11.4% of the total inflow from the entire watershed. We also analysed the impact of the stream flows in The ungauged zone onstreamflow significantly improves the water balance between inflow and outflow of the lake. These results, incorporating the estimated stream flow in ungauged zone, significantly improved the water balance as indicated by R2-with higher value and percent bias with lower value, as compared to the results when the stream flows in the ungauged zone were not taken into account, R2 with lower value and percent bias with higher value. the closing error decreased by 13.48 billion m3/a (10.10% if the total annual water resource) from 30.20 billion m3/a (20.10% of the total annual water resource) to 16.72 billion m3/a (10.00% of the total annual water resource).

The method can be extended to other lake, river, or ocean basins where stream flowstreamflow observation data is are unavailable, thus producing relatively accurate stream flowreasonable streamflow simulation results in ungauged zones. Reliable stream flowstreamflow 
[revised manuscript text omitted]

| Gauging-         | IndexInflow | Model Calibration (Jan.2000 Dec.2005)Streamflow set | Model Validation    |
|------------------|-------------|-----------------------------------------------------|---------------------|
| StationScenarios | Points      | at different points                                 | (Jan.2006 Dec.2011) |

| Original | d1  | $\mathbb{R}^2$ Od1: the observed streamflow at                                        | <del>Ens</del>                                                        | PBIAS(%)                       | ₽ 2 | <del>Ens</del> | PBIAS(%) |  |  |
|-----------------|------------|---------------------------------------------------------------------------------------|-----------------------------------------------------------------------|--------------------------------|-----------------------|----------------|----------|--|--|
| Scenario        |            | the Qiujin station (Q gau,d1 )                                             |                                                                       |                                |                       |                |          |  |  |
|                 | d2  | Od2: 50% of the observed streamflow                                                   | at the V                                                              | Vanjiabu station               | (Qgau,d2              | )              |          |  |  |
|                 | d3  | Od3: 10% of the observed streamflow                                                   | Od3: 10% of the observed streamflow at the Wanjiabu station (Qgau,d3) |                                |                       |                |          |  |  |
|                 | d4  | Od4: 20% of the observed streamflow                                                   | Od4: 20% of the observed streamflow at the Wanjiabu station (Qgau.d4) |                                |                       |                |          |  |  |
|                 | d5  | Od5: 20% of the observed streamflow                                                   | Od5: 20% of the observed streamflow at the Wanjiabu station (Qgau,d5) |                                |                       |                |          |  |  |
|                 | d6  | Od6: the observed streamflow at the L                                                 | .ijiadu s                                                             | tation (Q gau,d6 )  |                       |                |          |  |  |
|                 | d7  | Od7: the observed streamflow at the M                                                 | l eigang                                                       | station (Q gau,d7 ) |                       |                |          |  |  |
|                 | d8  | Od8: the observed streamflow at the H                                                 | Iushan s                                                       | station (Q gau,d8 ) |                       |                |          |  |  |
|                 | d9  | Od9: the observed streamflow at the D                                                 | Dufengk                                                               | eng station (Q gau  | (eb.           |                |          |  |  |
| Adjusted        | d1  | Ad1: the summation of Q ungau,d1 and Q                                     | Ad1: the summation of $Q_{ungau.d1}$ and $Q_{gau.d1}$                 |                                |                       |                |          |  |  |
| Scenario        | d2  | $\underline{d2} \qquad \underline{Ad2: the summation of Q_{ungau,d2} and Q_{gau,d2}}$ |                                                                       |                                |                       |                |          |  |  |
|                 | d3  | Ad3: the summation of Qungau, d3 and Q                                                | gau,d3                                                                |                                |                       |                |          |  |  |
|                 | d4  | Ad4: the summation of Qungau, d4 and Q                                                | gau,d4                                                                |                                |                       |                |          |  |  |
|                 | d5  | Ad5: the summation of Q ungau,d5 and Q                                     | Ad5: the summation of Q ungau,d5 and Q gau,d5   |                                |                       |                |          |  |  |
|                 | d6  | Ad6: the summation of Q ungau,d6 and Q gau,d6                   |                                                                       |                                |                       |                |          |  |  |
|                 | d7  | Ad7: the summation of Q ungau,d7 and Q gau,d7                   |                                                                       |                                |                       |                |          |  |  |
|                 | d8  | Ad8: the summation of Qungau, d8 and Q                                                | gau,d8                                                                |                                |                       |                |          |  |  |
|                 | d9  | Ad9: the summation of Qungau, d9 and Q                                                | gau,d9                                                                |                                |                       |                |          |  |  |
|                 | d10 | Ad10: Qungau,d10                                                                      |                                                                       |                                |                       |                |          |  |  |
|                 | d11 | Ad11: Qungau,d11                                                                      |                                                                       |                                |                       |                |          |  |  |

Table 2. The ungauged streamflow allocated to the lake inflow points of the dynamic model in the Adjusted Scenario. Qungau,di represent the ungauged streamflow gathering to the inflow point of di. d1, d2, d3... d11 are the inflow points in the Delft3D model and the outlets in the SWAT model (Fig. 1(b) and Fig. 3). b1, b2, b3...b11 are the subbasins in the PLUZ (Fig. 3(b)). Qswat,di represent the simulated discharges at the outlet (di) from the SWAT model. Qswat,Qiujin, Qswat,Wanjiabu, Qswat,Waizhou, Qswat,Lijiadu, Qswat,Meigang, Qswat,Hushan, and Qswat,Dufengkeng represent the simulated discharges at the outlets of Qiujin, Waizhou, Lijiadu, Meigang, Hushan and Dufengkeng respectively, from the SWAT model.

| the lake inflow point         | the subbasins draining to d i | the ungauged streamflow gathering to di                                     |
|-------------------------------|------------------------------------------|-----------------------------------------------------------------------------|
| (di) |                                          |                                                                             |
| d1                     | b12, b13 and b14                  | Qungau,d1: Qswat,d1- Qswat,Qiujin- Qswat,Wanjiabu                           |
| d2                     | b11                               | Qungau,d2: Qswat,d2- 50% *Qswat,Waizhou                                     |
| d3                     | b10                               | Qungau,d3: Qswat,d3- 10% *Qswat,Waizhou                                     |
| d4                     | b9                                | Qungau,d4: Qswat,d4- 20% *Qswat,Waizhou                                     |
| d5                     | b8                                | Qungau,d5 : Q swat,d5 - 20% *Q swat,Waizhou    |
| d6                     | b7                                | Qungau,d6: Qswat,d6- Qswat,Lijiadu                                          |
| d7                     | b6                                | Qungau,d7: Qswat,d7- Qswat,Meigang                                          |
| d8                     | b4 and b5                         | Qungau,d8: Qswat,d8- Qswat,Hushan                                           |
| d9                     | b3                                | Qungau,d9: Qswat,d9- Qswat,Dufengkeng                                       |
| d10                    | b2                                | Q ungau,d10 : Q swat,d10                              |
| d11                    | b1                                | Qungau,d11: Qswat,d11                                                       |
| total                         | b1, b2, b3, b4, b5, b6, b7,       | Qungau,total:                                                               |
|                               | b8, b9, b10, b11                  | $\underline{(Q_{swat,d1}+Q_{swat,d2}+Q_{swat,d3}+Q_{swat,d4}+Q_{swat,d5})}$ |

+Qswat,d6+Qswat,d7+Qswat,d8+Qswat,d9 +Qswat,d10+Qswat,d11)-(Qswat,Qiujin+Qswat,Wanjiabu+Qswat,Waizhou +Qswat,Lijiadu+Qswat,Meigang+Qswat,Hushan +Qswat,Dufengkeng)

975

**Table 3. Quantitative Assessment of Calibration and Validation for SWAT Model.**

| Gauging    | Index             | Model Calibration (2000-2005) |            |                  | Model Validation (2006-2011) |      |                  |
|------------|-------------------|-------------------------------|------------|------------------|------------------------------|------|------------------|
| Station    | muex              | R2          | Ens | PBIAS (%) | R2         | Ens  | PBIAS (%) |
| Wanjiabu   | monthly discharge | 0.63                          | 0.61       | -0.2             | 0.78                         | 0.76 | 9.4              |
| Waizhou    | monthly discharge | 0.94                          | 0.93       | 3.2              | 0.95                         | 0.93 | 6.5              |
| Lijiadu    | monthly discharge | 0.84                          | 0.82       | -9.4             | 0.88                         | 0.85 | -16.8            |
| Meigang    | monthly discharge | 0.89                          | 0.89       | 1.1              | 0.91                         | 0.90 | 10.0             |
| Hushan     | monthly discharge | 0.81                          | 0.78       | 14.2             | 0.76                         | 0.75 | 13.9             |
| Dufengkeng | monthly discharge | 0.80                          | 0.80       | -4.7             | 0.83                         | 0.80 | 9.4              |

Table 24. Quantitative assessment of calibration and validation for streamflow simulation for the Delft3D model.

| Gauging
Station | Index                        | Origin                                                | al Scenario        |                                                             |           |                        |              | Adjus           | ted Scenario |
|----------------------------------|------------------------------|--------------------------------------------------------------|--------------------|-------------------------------------------------------------|-----------|------------------------|--------------|------------------------|--------------|
| Gauging
Station               | Index                        | Calibration ( <del>Jan.</del> 2001-
<del>Dec.</del> 2005) |                    | Validation ( <del>Jan.</del> 2006-
<del>Dec.</del> 2010) |           | All (2001-2010) |              | All (2001-2010) |              |
|                                  |                              | $\mathbb{R}^2$                                               | PBIAS (%)          | $\mathbb{R}^2$                                              | PBIAS (%) | R2   | PBIAS (%)    | R2   | PBIAS (%)    |
| Xingzi                           | Lake water
level          | 0.99                                                         | 1. <del>20</del> 2 | 0.99                                                        | 0.45      | 0.99            | 0.85  | 0.99            | 0.48  |
| Duchang                          | Lake water
level          | 0.97                                                         | 4.74               | 0.99                                                        | 2.78      | 0.97            | 3.18  | 0.97            | 2.67         |
| Kangshan                         | Lake water
level          | 0.85                                                         | 2.86               | 0.88                                                        | 1.72      | 0.86            | 1.56  | 0.86            | 1.21  |
| Hukou                            | Lake
outflow
discharge | 0.75                                                         | 19.46              | 0.80                                                        | 21.47     | 0.77            | 20.10 | 0.81            | 10.00 |

980

Table 3. Annual water yields produced by the PLUZ (QPLUA) from 2000 to 2009. The table includes the whole Poyang Lake

985 catchment (Qwhole), and the ratio between QPLUA and Qwhole-

| Year            | $Q_{PLUA}(10^8 \text{m}^3)$ | $Q_{\text{whole}}(10^8 \text{m}^3)$ | QPLUA/Qwhole(%)   |
|-----------------|-----------------------------|-------------------------------------|-------------------|
| <del>2000</del> | <del>157.18</del>           | <del>1421.28</del>                  | <del>11.06%</del> |

| <del>2001</del> | <del>141.74</del> | <del>1477.88</del> | <del>9.59%</del>  |  |
|-----------------|-------------------|--------------------|-------------------|--|
| <del>2002</del> | <del>216.10</del> | <del>1856.29</del> | <del>11.64%</del> |  |
| <del>2003</del> | <del>220.90</del> | <del>1404.69</del> | <del>15.73%</del> |  |
| <del>2004</del> | <del>113.95</del> | <del>921.54</del>  | <del>12.36%</del> |  |
| <del>2005</del> | <del>187.83</del> | <del>1471.95</del> | <del>12.76%</del> |  |
| <del>2006</del> | <del>155.76</del> | <del>1560.27</del> | <del>9.98%</del>  |  |
| <del>2007</del> | <del>72.41</del>  | <del>1012.19</del> | <del>7.15%</del>  |  |
| <del>2008</del> | <del>133.71</del> | <del>1291.85</del> | <del>10.35%</del> |  |
| <del>2009</del> | <del>115.70</del> | <del>1057.66</del> | <del>10.94%</del> |  |
| The Average     | <del>151.53</del> | <del>1347.56</del> | <del>11.24%</del> |  |

**Figures**

---

## Referee Report (RR1)

Comments on "Stream flow simulation and verification in ungauged zones by coupling hydrological and hydrodynamic models: a case study of the Poyang Lake ungauged zone" by L. Zhang et al.

The authors improved the manuscript and the methodology is clear for me now.    However, there are still some obvious typos, errors and some statements are still unclear.    What's more, I still remain my concern about the significance of this study and the results.    The purpose of this study is to verify the stremflow simulation in ungauged zones surrounding the large water bodies by coupling hydrological and hydrodynamic models.    However, I don't think the current results could achieve this goal because the error for the adjusted scenario is still as high as about 10% (Lines 23-25).    How do authors prove this 10% error is not caused by the bad performance of SWAT model in the ungagged zone?    Therefore, I still cannot recommend publication in HESS for this version of the manuscript and my specific comments are listed below:

1.  When authors address reviewer's comments, I highly recommend authors highlight the revised paragraph for addressing each comment.    It's more professional and convenient for reviewer to judge whether each comment has been addressed.    For example:
    Reviewer's comment (one paragraph)
    Authors' response (one or more paragraphs)
    Revised paragraph of manuscript (Lines***-****): "paste the revised paragraph or sentences"
2.  Are the parameters of hydrodynamic model for the adjusted and original scenarios the same? If yes, how to set up/calibrate the parameters? If no, why?
3.  How the water yield is defined or computed in this study? The water yield is defined as the difference between precipitation and evapotranspiration. It's not the equivalence of the streamflow, especially for short time period.
4.  Lines 240 – 242: gramma error.    Please double check.
5.  Lines 246 – 247: I don't agree with this statement.    If the simulated streamflow is wrong or the performance of SWAT is bad in ungauged zone, I think the $\varepsilon_{adj}$ should include the uncertainties in the simulated streamflow in the ungauged zone.
6.  Caption of Figure 8, please clarify the annual water yield from which part of the study area.
7.  Line 292: Replace "Fig.8 (b)" with "Fig. 8".

---

## Author Response (AR2)

Replies to Referee Report 2

We are very grateful to the reviewer for reading the manuscript extremely carefully and forwarding the valuable suggestions for improvement. Point-by-point responses to the reviewers' comments are listed below.

**1.**

**Reviewer's general comment:** The purpose of this study is to verify the streamflow simulation in ungauged zones surrounding the large water bodies by coupling hydrological and hydrodynamic models. However, I don't think the current results could achieve this goal because the error for the adjusted scenario is still as high as about 10% (Lines 23-25). How do authors prove this 10% error is not caused by the bad performance of SWAT model in the ungagged zone?

**Authors' response:** Thank you for the good suggestion. It is difficult to *prove the 10% error is not caused by bad performance of the SWAT model in the ungauged zone*. So it is hard to directly verify the performance of the SWAT model in the ungauged zone. Therefore we use an indirect way to archive the verification purpose. We compare the results of the two lake hydrodynamic model scenarios (Adjusted and Original Scenarios). Adjusted Scenario considers the SWAT simulated ungauged streamflow, while Original Scenario does not. The improved result indicates the ungaued streamflow is reasonable. I think it indeed can verify the ungauged streamflow in an indirect way.

**Revised paragraph of manuscript:** no revised words.

**2.**

**Reviewer's general comment 1:** When authors address reviewer's comments, I highly recommend authors highlight the revised paragraph for addressing each comment.    It's more professional and convenient for reviewer to judge whether each comment has been addressed.    For example:

Reviewer's comment (one paragraph)

Authors' response (one or more paragraphs)

Revised paragraph of manuscript (Lines***-****): "paste the revised paragraph or sentences"

**Authors' response:** Thank you very much for the suggestion.

**Revised paragraph of manuscript:** no revised words.

**3.**

**Reviewer's general comment 2:** Are the parameters of hydrodynamic model for the adjusted and original scenarios the same? If yes, how to set up/calibrate the parameters? If no, why?

**Authors' response:** Yes, the parameter in the two scenarios are the same.

We set the parameter(the Manning roughness coefficient, the eddy viscosity parameter and the critical water depth for wetting and drying) as the fittest ones calibrated by Zhang et al. (2015) as we applies the same hydrodynamic model (Delft3D) in the same study area (Poyang Lake). Zhang et al. use try-and-error method to calibrate these parameter. The fittest parameter is firstly applied to the model in Original Scenario ensuring that the parameter is reasonable for the application. In fact, for calibration and validation duration in Original Scenario, the simulated results meet a good performance (ENS>0.8, absolute PBIAS<3% for water level and ENS>0.7, absolute PBIAS<22% for streamflow). Then we use the same parameter for Adjusted Scenario.

The reference: Zhang, P., Lu, J., Feng, L., Chen, X., Zhang, L., Xiao, X., & Liu, H. Hydrodynamic and

inundation modeling of China's largest freshwater lake aided by remote sensing data. Remote Sensing, 7, 4858-4879, doi: 10.3390/rs70404858, 2015.

**Revised paragraph of manuscript (Line 184 - 185):**

**Before the revises:**

Two scenarios were established, the adjusted scenario (Adjusted Scenario) and the original scenario (Original Scenario).

Original Scenario did not take streamflow in the PLUZ into consideration, unlike Adjusted Scenario, which accounted for the ungauged zones

**After the revises:**

Two scenarios were established, the adjusted scenario (Adjusted Scenario) and the original scenario (Original Scenario). We applies the same hydrodynamic model (Delft3D) in the same study area (Poyang Lake) as the research by Zhang et al (2015). Therefore, we set the parameter (the Manning roughness coefficient, the eddy viscosity parameter and the critical water depth for wetting and drying) as the fittest ones calibrated by Zhang et al. for Original Scenario. The parameters in the two scenarios are the same.

Original Scenario did not take streamflow in the PLUZ into consideration, unlike Adjusted Scenario, which accounted for the ungauged zones.

**4.**

**Reviewer's general comment 3:** How the water yield is defined or computed in this study? The water yield is defined as the difference between precipitation and evapotranspiration. It's not the equivalence of the streamflow, especially for short time period.

**Authors' response:** The water yield is the accumulative streamflow in a specified duration. Monthly water yield is the accumulative streamflow in a specified month. Annual water yield is the accumulative streamflow in a specified year. In the paper, the units of streamflow, monthly water yield and annual water yield are $m^3/s$, $m^3/month$ and $m^3/a$ respectively.

**Revised paragraph of manuscript (Line 223 - 224)**: Add the following paragraph between Line 223 – 224.

In a duration time, water yield can reflect the total amount. So we analysis the water yield variable instead of streamflow. Water yield is computed as the accumulative streamflow in a specific duration. Monthly water yield is the accumulative streamflow in a specified month. Annual water yield is the accumulative streamflow in a specified year. In the paper, the units of streamflow, monthly water yield and annual water yield are $m^3/s$, $m^3/month$ and $m^3/a$ respectively.

**5.**

**Reviewer's general comment 4:** Lines 240 – 242: gramma error. Please double check.

**Authors' response:** Thank you very much for the suggestions.

**Revised paragraph of manuscript (Line** 240 – 242**):**

**Before the revises:**

As the ungauged zone occupies 12% of the total water balance (Li et al. 2014), much larger than the other components (E and G, less than 3.3%), the closing error should be large than zero if the observe data and hydrodynamic model are sufficient accuracy.

**After the revises:**

As the ungauged zone occupies 12% of the total water balance components (Li et al. 2014), much

larger than the other components (E and G, less than 3.3%), the closing error should be larger than zero on the assumption that the observe data and hydrodynamic model are sufficient accuracy.

**6.**

**Reviewer's general comment 5:** Lines 246 – 247: I don't agree with this statement. If the simulated streamflow is wrong or the performance of SWAT is bad in ungauged zone, I think the $\varepsilon_{adj}$ should include the uncertainties in the simulated streamflow in the ungauged zone.

**Authors' response:** Thank you very much for the suggestion. The expression is not proper. $\varepsilon_{adj}$ does include the uncertainty arising from the SWAT model for the ungauged zone.

$\varepsilon_{org}$ represents the uncertainty in Original Scenario, which arising from the ignorance of the ungauged streamflow, E, G, the error in the observe data, and uncertainty of the hydrodynamic model. $\varepsilon_{adj}$ represents the uncertainty in Adjusted Scenario, which arising from the ignorance of E, G, the error in the observe data, and uncertainty of the hydrodynamic model and the simulated ungauged streamflow result. However, the partial uncertainties (caused by the ignorance of E, G, the error in the observe data, and uncertainty of the hydrodynamic model) in Adjusted Scenario and Original Scenario are the same. Therefore, if the simulated ungauged stream by the SWAT model are sufficient accuracy, the uncertainty in Adjusted Scenario ($\varepsilon_{adj}$) should be smaller than that in Original Scenario ($\varepsilon_{org}$). Conversely, if the uncertainty in Adjusted Scenario ($\varepsilon_{adj}$) is smaller than that in Original Scenario ($\varepsilon_{org}$), we can demonstrate that the simulation result of the ungauged streamflow by the SWAT model is reasonable, verifying the simulated ungauged streamflow in an indirect way.

**Revised paragraph of manuscript (Line 238 - 240):**

   **Before the revises:**

$\varepsilon_{org}$ represents the uncertainty of the equation, which arising from the ungauged streamflow, E, G, the error in the observe data, and uncertainty of the hydrodynamic model.

   **After the revises:**

$\varepsilon_{org}$ represents the uncertainty of the equation, which arising from the ignorance of the ungauged streamflow, E, G, the error in the observe data, and uncertainty of the hydrodynamic model.

**Revised paragraph of manuscript (Line 246 - 247):**

   **Before the revises:**

$\varepsilon_{adj}$ represents the uncertainty, which doesn't include the ungauged streamflow. Thus, the absolute value of $\varepsilon_{adj}$ should be smaller than that of $\varepsilon_{org}$, if the observe data and hydrodynamic model are sufficient accuracy.

   **After the revises:**

$\varepsilon_{adj}$ represents the uncertainty of the equation, which arising from the ignorance of E, G, the error in the observe data, and uncertainty of the hydrodynamic model and the simulated ungauged streamflow result. The partial uncertainties (caused by the ignorance of E, G, the error in the observe data, and uncertainty of the hydrodynamic model) in Adjusted Scenario and Original Scenario are the same. Thus, if the simulated ungauged stream by the SWAT model are sufficient accuracy, the uncertainty in Adjusted Scenario ($\varepsilon_{adj}$) should be smaller than that in Original Scenario ($\varepsilon_{org}$).

**7.**

**Reviewer's general comment 6:** Caption of Figure 8, please clarify the annual water yield from

which part of the study area.

**Authors' response:** The annual water yield is from the ungauged zone. That is the accumulative streamflow simulated by the SWAT model in the ungauged zone.

**Revised paragraph of manuscript (Line 494 - 495):**

   **Before the revises:**

Figure 8. The variation trend of the annual water yield from 2001 to 2009. It shows declining trend at a rate -1.02 of billion m3/a (P < 0.05).

   **After the revises:**

Figure 8. The variation trend of the annual water yield of the ungauged zone from 2001 to 2009. It shows declining trend at a rate -1.02 of billion m3/a (P < 0.05).

**Revised paragraph of manuscript (Line 310):**

   **Before the revises:**

Annual streamflow shows a clear declining trend (P<0.05, from t-test), at a rate of -1.02 billion m3/a (dashed line in Fig. 8) during the period from 2001 to 2009.

   **After the revises:**

Annual streamflow of the ungauged zone shows a clear declining trend (P<0.05, from t-test), at a rate of -1.02 billion m3/a (dashed line in Fig. 8) during the period from 2001 to 2009.

**8.**

**Reviewer's general comment 7:** Line 292: Replace "Fig.8 (b)" with "Fig. 8".

**Authors' response:** Thank you very much for the suggestion.

**Revised paragraph of manuscript (Line 292):**

   **Before the revises:**

[revised manuscript text omitted]